# Critic–Adviser–Reviser Cyclic Refinement: Towards High-Quality EMR Corpus Generation with LLMs

 Chen Ning[1], Xien Liu[*1], Chenwei Yan[2], Xiao Zhang[3], Xinxin You[1], Yuxuan Zhou[1],
Xiangling Fu[4,5], Ji Wu[1,6,7]
[1]Department of Electronic Engineering, Tsinghua University
[2]School of Artificial Intelligence and Data Science,
University of International Business and Economics        [3]ByteDance
[4]School of Computer Science (National Demonstrative Software School),
Beijing University of Posts and Telecommunications
[5]Key Laboratory of Trustworthy Distributed Computing and Service (BUPT),
Ministry of Education        [6]College of AI, Tsinghua University        [7]BNRist

## Abstract

Electronic medical records (EMRs) are vital for healthcare research, but their use is limited by privacy concerns. Synthetic EMR generation offers a promising alternative, yet most existing methods merely imitate real records without adhering to rigorous clinical quality principles. To address this, we introduce LLM-CARe, a stage-wise cyclic refinement framework that progressively improves EMR quality through three stages, each targeting a specific granularity: corpus, section and document. At each stage, a Critic, an Adviser, and a Reviser collaborate iteratively to evaluate, provide feedback, and refine the drafts. This structured, multi-stage process produces records that better satisfy clinical quality standards. Experiments show that LLM-CARe significantly enhances EMR quality across all levels compared to strong baselines and yields improved performance on real-world clinical tasks such as diagnosis prediction. Unlike prior work, our method requires no real EMR text for training or prompting, demonstrating the effectiveness of stage-wise, cyclic refinement for generating high-quality, privacy-preserving EMR datasets.

## 1 Introduction

Electronic Medical Records (EMRs) are a valuable resource for healthcare research (Ma et al., 2017; Shang et al., 2019; Shen et al., 2025), offering large-scale, clinically grounded insights that reflect real-world medical practice. However, the sensitive nature of patient information poses significant privacy challenges, which severely limit the open sharing and use of real EMRs (Iyengar et al., 2018; Shah & Khan, 2020; Tertulino et al., 2024). To mitigate these concerns, researchers have explored synthetic EMR generation methods that aim to preserve data utility while protecting patient confidentiality (Yan et al., 2022; Murtaza et al., 2023; Yuan et al., 2024; Lin et al., 2025).

Existing EMR synthesis approaches primarily focus on mimicking real records (Lee, 2018; Baowaly et al., 2018; Yoon et al., 2023), without explicit ensuring clinical soundness. This imitation-based strategy is risky: real EMRs may contain errors (Aerts et al., 2021; Mohamed et al., 2023), which can be inadvertently inherited by synthetic data (Figure 1(a)). In practice, EMRs are professional medical documents whose reliability depends on satisfying key requirements such as completeness, consistency, and distribution alignment. Synthetic records that fail to meet such requirements may be less useful—or even misleading—for downstream clinical or research applications.

Recent advances in large language models (LLMs) make them a promising tool for EMR generation, due to their text generation ability and rich internal knowledge. However, as shown in Figure 1(a), our preliminary analysis reveals that LLM outputs often exhibit biased distributions—such as unrealistic gender patterns—and tend to produce only the most typical presentations of a disease, lacking

---
[*]Corresponding author.

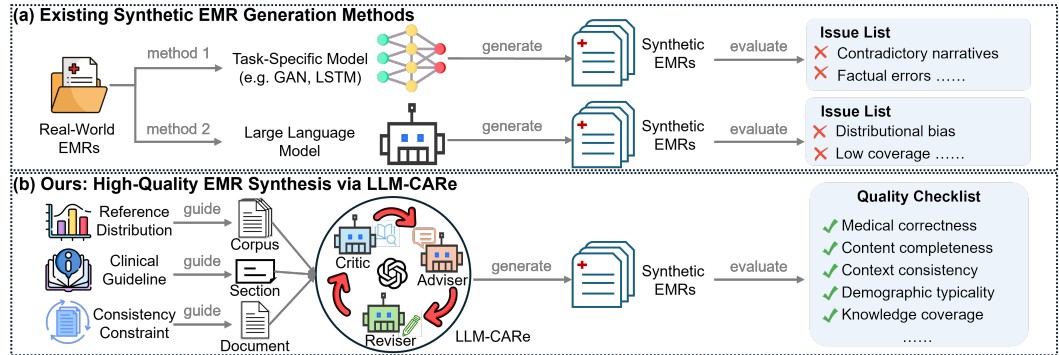

Figure 1: **(a)** Traditional EMR generation that mimic real EMRs without considering quality often leads to suboptimal outputs. **(b)** Our proposed LLM-CARe incorporates cyclic refinement based on quality principles, synthesizing high-quality EMRs.

coverage of the diverse and less common clinical scenarios seen in practice. These challenges highlight the need for more structured approaches to harness LLMs effectively for EMR synthesis.

To bridge this gap, we propose **LLM**-based **C**ritic–**A**dviser–**R**eviser Cyclic R**e**finement (**LLM-CARe**), a framework that enhances synthetic EMR quality through progressive refinement across three levels. As illustrated in Figure 1(b), LLM-CARe incorporates clinical quality principles into the generation process, producing records that align with standards of medical documentation. These requirements are organized into concrete principles of corpus distributional alignment, section completeness, and document consistency, forming the basis for refinement. Guided by them, LLM-CARe proceeds through three stages of refinement—corpus, section, and document—each targeting a distinct aspect of EMR quality. Within every stage, a Critic, an Adviser, and a Reviser collaborate in a cyclic loop: the Critic evaluates the drafts, the Adviser provides targeted feedback, and the Reviser updates the records. While the interaction pattern is shared, the role of each agent adapts to the stage: corpus stage aligns the dataset with realistic distributions, section stage enforces section completeness, and document stage ensures logical consistency within record. This structured process enables systematic enhancement of EMRs from local detail to global corpus characteristics.

To validate the effectiveness of our approach, we conduct experiments on a real-world EMR dataset containing 192k records across 302 diseases. We assess the intrinsic quality of generated records using a strong LLM as a judge, complemented by statistical comparisons to real EMRs. Additionally, we evaluate downstream utility by training models on synthetic EMRs and testing on real records across three representative clinical tasks: diagnosis prediction, examination recommendation, and treatment recommendation. Results show that LLM-CARe consistently outperforms baselines in both record quality and task performance. Notably, our method requires no access to real EMR text for training or prompting during generation, fully preserving patient privacy while producing clinically meaningful and practically useful EMRs. Our main contributions are summarized as follows:

- We propose **LLM-CARe**, a stage-wise multi-agent framework for high-quality EMR synthesis that employs cyclic refinement based on clinical quality principles.
- LLM-CARe consistently improves EMR quality compared to baselines across multiple levels. Further analysis shows that all three agents play essential and complementary roles.
- Without training or prompting on real EMR text, our synthetic data yields superior downstream task performance compared to baselines, ensuring both utility and privacy.

## 2 RELATED WORK

There has been growing interest in synthesizing EMRs to address privacy concerns and facilitate secure data sharing. We categorize existing methods into three main paradigms:

**GAN-based EMR Generation.** Generative adversarial networks (GANs) have been extensively explored for EMR synthesis. Some works generate EMRs from random noise vectors (Choi et al.,

2017; Baowaly et al., 2018; Chin-Cheong et al., 2019; Yoon et al., 2023), while other methods incorporate structured conditions—such as diagnosis codes—to guide generation process (Rashidian et al., 2020; Zhang et al., 2020; Guan et al., 2021; Li et al., 2023). Although effective at modeling data distributions, these methods typically ignore the clinical quality of the generated records.

**Auto-regressive EMR Generation.** Another line of research leverages auto-regressive models to generate EMRs. Recurrent neural networks (RNNs) have been used to model sequential EMR data (Lee, 2018; Melamud & Shivade, 2019; Mosquera et al., 2023; Ganguli et al., 2023), and more recently, transformer-based architectures have been introduced to capture long-range dependencies within records (Wang et al., 2019; Amin-Nejad et al., 2020; Theodorou et al., 2023; Karami et al., 2024). While these models excel at learning temporal and structural patterns, they generally treat EMRs as sequences of tokens without mechanisms to ensure clinically meaningful coherence.

**LLM-based EMR Generation.** With the emergence of large language models (LLMs), recent studies have explored prompting LLMs to synthesize EMRs, either by providing brief clinical descriptions or by asking the model to emulate real patient records (Litake et al., 2024; Kumichev et al., 2024; Abdel-Khalek et al., 2024; Kweon et al., 2024; Lin et al., 2025). While LLMs exhibit strong capabilities, direct generation often results in diverge from realistic corpus-level distributions.

Recent work has also explored diffusion models for EMR synthesis (Yuan et al., 2023), but existing methods primarily target numerical representations rather than free-text EMRs, making them less applicable to the textual EMR generation setting studied in this work.

## 3 METHOD

In this section, we present **LLM-CARe**, a stage-wise cyclic refinement framework for enhancing the quality of synthetic EMRs. Unlike ordinary free-form text, EMR must satisfy quality requirements to be clinically meaningful. We focus on five principles across three levels: **demographic typicality** (plausible population patterns), **knowledge coverage** (sufficient breadth of clinical conditions), **content completeness** (each section includes its essential information), **medical correctness** (information is clinically valid), and **context consistency** (sections do not contradict one another). Each principle is further

| Level | Principle | Example of Criteria |
|---|---|---|
| Corpus | Demographic Typicality | The gender ratio of synthetic data should align with real populations. |
| | Knowledge Coverage | The synthetic dataset should cover comprehensive symptoms. |
| Section | Content Completeness | The chief complaint should state the reason for admission. |
| Document | Medical Correctness | The diagnosis should be valid given the patient's gender. |
| | Context Consistency | Symptoms in chief complaint and history of present illness should align. |

Figure 2: Quality principles across three levels with representative criteria.

divided into concrete criteria that can be explicitly checked and refined. Figure 2 shows representative examples, with full details in Appendix A.

To address these requirements, LLM-CARe refines EMRs in three successive stages—corpus, section, and document. As illustrated in Figure 3, each stage involves the collaboration of three agents in a cyclic loop: the *Critic*, who evaluates drafts against stage-specific objectives; the *Adviser*, who pinpoints areas for improvement and suggests strategies; and the *Reviser*, who incorporates feedback to update the drafts. This iterative process progressively improves EMRs from global distributional alignment, to field-level completeness, and finally to record-level coherence.

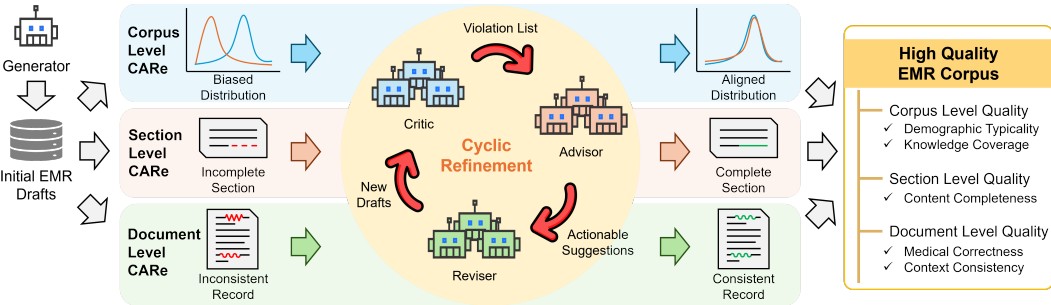

Figure 3: Overview of our proposed LLM-CARe framework for synthesizing high-quality EMRs.

## 3.1 INITIAL DRAFT GENERATION

The generation process begins with a *generator* agent $\mathcal{M}_{\text{generator}}$, which produces initial EMR drafts based on an input prompt $x$. This prompt specifies key information such as the target primary diagnosis and required EMR fields (e.g., chief complaint, history of present illness). For each prompt, the generator samples multiple drafts to form a starting draft pool:

$$\mathcal{D}^{(0)} = \{E_1^{(0)}, \ldots, E_n^{(0)}\}, \quad E_i^{(0)} \sim \mathcal{M}_{\text{generator}}(x) \tag{1}$$

where $\mathcal{D}^{(0)}$ denotes the initial draft set and each $E_i^{(0)}$ is an EMR instance. These drafts may exhibit omissions, inconsistencies, or clinically implausible details, but they provide the foundation for subsequent stage-wise refinement.

At the dataset scale, high-quality synthetic EMRs must preserve realistic and representative distributions. We focus on two aspects: **Demographic Typicality**, ensuring variables such as age and gender reflect real-world patient populations (see Figure 11 for examples), and **Knowledge Coverage**, ensuring the corpus contains both common and rare clinical conditions (with detailed dimensions listed in Table 9). Maintaining these corpus-level properties is essential, as deviations may introduce demographic bias or insufficient coverage of clinically important but less frequent scenarios. These goals are addressed through corpus-level agent interactions, where the critic, adviser, and reviser collaborate to align the overall distribution.

*Corpus-level Critic.* At the corpus stage, the critic focuses on dataset-wide properties, capturing how well synthetic EMRs aligns with target distributions. For each attribute $c_{\text{corpus},k}$ (e.g., age, gender, or a knowledge dimension), it measures the deviation of the current corpus $\mathcal{D}^{(t)}$ from the reference distribution $\mathcal{T}_d$ derived from aggregate statistics on the training set, including age and gender distributions and the frequencies of clinical concepts of interest (e.g., symptoms, medications):

$$\delta_{\text{corpus},k}^{(t)} = \mathcal{M}_{\text{critic}}^{\text{corpus}}\left(\mathcal{D}^{(t)}, \mathcal{T}_d, c_{\text{corpus},k}\right) \tag{2}$$

*Corpus-level Adviser.* The adviser interprets the critic's feedback to guide modifications at the dataset level. Based on the deviations, it identifies a subset of records $\mathcal{S}_k^{(t)} \subset \mathcal{D}^{(t)}$ whose adjustment would most effectively reduce distributional mismatch, and generates actionable feedback $F_{\text{corpus},k}^{(t)}$:

$$\mathcal{S}_k^{(t)}, F_{\text{corpus},k}^{(t)} = \mathcal{M}_{\text{adviser}}^{\text{corpus}}\left(\mathcal{D}^{(t)}, c_{\text{corpus},k}, \delta_{\text{corpus},k}^{(t)}\right) \tag{3}$$

*Corpus-level Reviser.* The reviser applies the adviser's instructions to the selected subset $\mathcal{S}_k^{(t)}$, modifying or enriching records to better match the reference distribution:

$$\mathcal{S}_k^{(t+1)} = \mathcal{M}_{\text{reviser}}^{\text{corpus}}\left(\mathcal{S}_k^{(t)}, F_{\text{corpus},k}^{(t)}\right) \tag{4}$$

## 3.2 SECTION-LEVEL CARE

At the section scale, high-quality EMRs must ensure **Content Completeness**: each field should contain the essential clinical elements expected for its type. Ensuring section-level completeness is crucial, as omissions in key fields can result in records that misrepresent the patient's condition and compromise clinical validity. To operationalize this, we define a set of section-specific criteria derived from clinical guidelines (see Table 6), and apply cyclic refinement with critic, adviser, and reviser agents to supplement missing information.

*Section-level Critic.* The critic operates for each single section. For a section $s_{i,m}^{(t)}$ in record $E_i^{(t)}$ and a criterion $c_{\text{sec},k}$ derived from clinical guidelines, it determines whether the criterion is met:

$$\delta_{\text{sec},i,k}^{(t)} = \mathcal{M}_{\text{critic}}^{\text{sec}}\left(s_{i,m}^{(t)}, c_{\text{sec},k}\right), \quad \delta_{\text{sec},i,k}^{(t)} \in \{0, 1\} \tag{5}$$

*Section-level Adviser.* For unmet criteria ($\delta_{\text{sec},i,k}^{(t)} = 0$), the adviser examines the section and designs specific instructions to indicate exactly which clinical elements should be added or clarified:

$$F_{\text{sec},i,k}^{(t)} = \mathcal{M}_{\text{adviser}}^{\text{sec}}\left(s_{i,m}^{(t)}, c_{\text{sec},k}\right) \tag{6}$$

*Section-level Reviser.* Using the adviser's guidance, the reviser updates the section by incorporating the recommended elements while preserving existing content and coherence:

$$s_{i,m}^{(t+1)} = \mathcal{M}_{\text{reviser}}^{\text{sec}}\left(s_{i,m}^{(t)}, F_{\text{sec},i,k}^{(t)}\right) \tag{7}$$

Through this cycle, sections are progressively completed and made sufficient for their clinical role.

## 3.3 DOCUMENT-LEVEL CARE

At the document scale, EMRs must ensure both **Medical Correctness**—that clinical statements are valid given the diagnosis—and **Context Consistency**—that information across sections does not conflict. Without enforcing these criteria, even small cross-section inconsistencies or incorrect clinical relations can undermine the internal logic of the record to the point where the overall EMR becomes clinically unreliable. To make these requirements concrete, we define detailed criteria for both aspects (see Tables 7 and Table 8). To enforce them, we refine EMRs through document-level agent interactions, where the focus is on coherence across multiple sections.

*Document-level Critic.* The critic evaluates each record as a whole, checking constraints across sections for logical and clinical plausibility. For a consistency criterion $c_{\text{doc},k}$, it outputs a judgment:

$$\delta_{\text{doc},i,k}^{(t)} = \mathcal{M}_{\text{critic}}^{\text{doc}}\left(E_i^{(t)}, c_{\text{doc},k}\right), \quad \delta_{\text{doc},i,k}^{(t)} \in \{0,1\} \tag{8}$$

*Document-level Adviser.* When inconsistencies are flagged ($\delta_{\text{doc},i,k}^{(t)} = 0$), the adviser generates targeted feedback, often suggesting edits to the less influential section to restore harmony:

$$F_{\text{doc},i,k}^{(t)} = \mathcal{M}_{\text{adviser}}^{\text{doc}}\left(E_i^{(t)}, c_{\text{doc},k}\right) \tag{9}$$

*Document-level Reviser.* Finally, the reviser integrates this feedback to harmonize the conflicting sections and yield an updated document:

$$E_i^{(t+1)} = \mathcal{M}_{\text{reviser}}^{\text{doc}}\left(E_i^{(t)}, F_{\text{doc},i,k}^{(t)}\right) \tag{10}$$

Through this process, records are refined into coherent, consistent, and clinically valid narratives.

## 3.4 STAGE-WISE ORDERING

At each stage, the critic, adviser, and reviser interact in cycles to refine the drafts according to stage-specific principles. Once the drafts have been improved at the current granularity, they are passed to the next stage, where the agent interaction continues under a different focus. The staged order is intentional: modifications at one level can influence others, so we proceed **from the most flexible to the most stringent stage**. Corpus-level refinement is relatively soft, aiming to align distributions without requiring exact matches, and is therefore performed first. While document-level refinement enforces strict logical consistency across sections, where errors could introduce serious contradictions, thus is performed last. By progressing in this order, each stage builds on the previous one while minimizing unintended conflicts. Through this staged refinement, the synthetic EMRs achieve high quality across corpus, section, and document levels.

## 4 EXPERIMENTAL SETUP

**Dataset** To validate the effectiveness of our method, we conduct experiments on a real-world EMR dataset comprising 192k records across 302 diseases. The dataset is carefully de-identified by removing all sensitive patient information. Unlike many prior studies that focus on synthesizing a single field in isolation (e.g., chief complaint), we consider multiple fields that together capture the clinical trajectory from admission to discharge to provide a comprehensive view of each clinical episode. To ensure consistent disease distribution across subsets, we perform an 8:2 stratified split based on disease categories, maintaining proportional representation in both the training and test sets. Further details are provided in Appendix B.

**Baselines**   We compare our method against two categories of baselines. **(1) Non-LLM generative models. LSTM** (Lee, 2018) and **mtGAN** (Guan et al., 2021) share the same generation setup—both take the disease label as input and are trained from scratch on the training split to produce full EMRs, differing only in model architecture (LSTM vs. GAN). **(2) LLM-based generators. MedSyn** (Kumichev et al., 2024) generates EMRs by prompting an LLM with real EMR examples from the training set along with realted symptoms; **LLM Direct** generates EMRs from instructions without quality control; and **Self-Refine** (Madaan et al., 2023) performs iterative self-critique that evaluates all quality requirements jointly in each refinement round. None of these LLM-based baselines involve fine-tuning. All approaches are conditioned on disease labels, and generate the same number of EMRs with same disease distribution as the test set. Details are provided in Appendix C.

**Evaluation Settings**   We employ two types of metrics to evaluate synthetic EMRs:

**EMR quality** is assessed based on the five principles introduced above. For medical correctness, content completeness, and context consistency, we use an LLM-based judge that evaluates each EMR against our criterion set. For demographic typicality and knowledge coverage, we compute statistical similarity between synthetic and real data within each disease—comparing age and gender distributions, and measuring the proportion of clinically relevant entities (symptoms, exams, treatments) covered in synthetic EMRs.

**Downstream task performance** provides an practical way to evaluate the utility of synthetic data. Following prior work, we train task-specific models on synthetic EMRs and evaluate their performance on real-world test data. This setup reflects a common use case of synthetic data in low-resource scenarios. We consider three representative tasks of high clinical relevance: diagnosis prediction, examination recommendation, and treatment recommendation. These tasks collectively span key aspects of medical decision-making. Each task is framed as a multiple-choice question, where the model predicts answers based on the chief complaint and history of present illness.

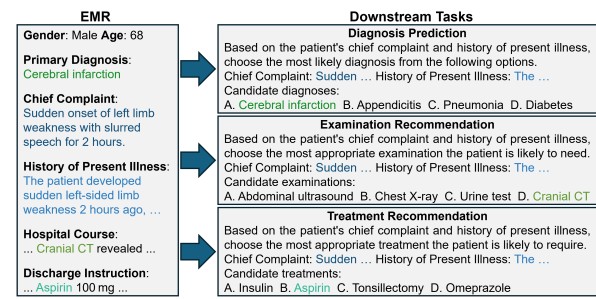

Figure 4: Construction of multiple-choice questions for three downstream tasks from an EMR.

Illustrative examples are shown in Figure 4. More details are provided in Appendix D.

**Implementation Details**   We use Qwen2.5-7B-Instruct (Yang et al., 2025) as the backbone model for all LLM-based methods. For EMR quality evaluation, we adopt Qwen2.5-32B-Instruct, as larger models tend to provide more reliable judgments. For downstream tasks, we fine-tune Qwen2.5-0.5B-Instruct on synthetic EMRs and evaluate performance on real test data. Detailed experimental settings are provided in Appendix E.

## 5   EXPERIMENTAL RESULTS AND DISCUSSION

### 5.1   COMPARISON OF EMR QUALITY

Table 1 summarizes performance across the five quality principles. LLM-CARe consistently outperforms all baselines on every metric, improving both individual-record quality and corpus-level characteristics. Among the baselines, LLM-based baselines generally exceed traditional models on section- and document-level criteria, yet without structured refinement they can underperform on corpus-level dimensions. Self-Refine provides only minor gains over LLM Direct and remains far below our method, indicating that generic all-in-one revision cannot satisfy EMRs' multi-level quality requirements. This shows that **simply employing LLMs is not sufficient to guarantee comprehensive EMR quality**. In contrast, LLM-CARe achieves strong and balanced improvements across all levels through principle-guided, stage-wise refinement. To confirm that these findings are not tied to a specific judge, we also evaluate using additional LLMs (Appendix F). Fine-grained subgroup analyses across demographic and disease-frequency partitions are provided in Figure 12.

Table 1: Quality score (%) of generated EMRs across principles, where higher values indicate better performance. (*) denotes standard deviation calculated from 3 runs with different random seeds.

| Type | Method | Rely on EMR Text | Section Level | Document Level | | Corpus Level | |
|------|--------|:---:|:---:|:---:|:---:|:---:|:---:|
| | | | Content Completeness | Medical Correctness | Context Consistency | Demographic Typicality | Knowledge Coverage |
| Non-LLM | LSTM | ✓ | 70.8(1.1) | 65.0(0.4) | 21.7(2.3) | 93.3(0.6) | 70.4(0.4) |
| | mtGAN | ✓ | 55.8(2.9) | 51.8(6.2) | 21.4(4.2) | 93.6(1.4) | 76.3(3.7) |
| LLM-Based | MedSyn | ✓ | 84.8(0.3) | 95.3(0.8) | 91.9(1.1) | 84.1(0.9) | 84.5(5.8) |
| | LLM Direct | ✗ | 77.1(0.1) | 90.7(0.2) | 87.9(0.1) | 77.7(0.1) | 73.9(0.2) |
| | Self-Refine | ✗ | 78.3(0.2) | 90.9(0.0) | 88.5(0.1) | 77.7(0.4) | 78.0(2.4) |
| | **LLM-CARe(ours)** | ✗ | **91.2**(0.4) | **98.6**(0.0) | **93.8**(0.1) | **96.8**(1.4) | **94.1**(0.1) |

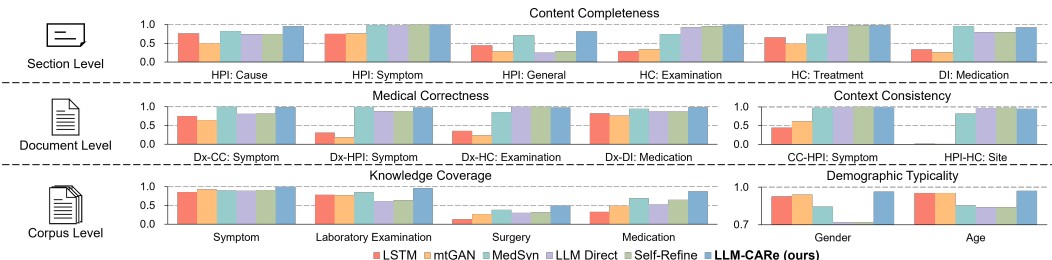

Figure 5: Detailed EMR quality evaluation across 3 levels. Abbreviations: CC-Chief Complaint, HPI-History of Present Illness, HC-Hospital Course, DI-Discharge Instructions, Dx-Diagnosis.

Figure 5 presents a fine-grained breakdown of EMR quality across several representative and clinically important criteria; complete results are provided in Appendix H. Our method achieves the best or competitive performance on the majority of criteria, demonstrating robustness across diverse quality dimensions. Notably, MedSyn underperforms even the LLM Direct on some criteria, such as the completeness of the hospital course. **This suggests that real EMRs—used by MedSyn as in-context exemplars—may contain omissions that propagate into the generated records.** These findings further highlight the limitations of purely imitative approaches and emphasize the importance of explicitly modeling and enforcing quality standards during generation.

## 5.2 COMPARISON OF DOWNSTREAM TASK PERFORMANCE

Table 2: Accuracy (%) of three downstream tasks, where micro and macro are averaged across diseases. (*) denotes standard deviation calculated from 3 runs with different random seeds.

| Type | Method | Rely on EMR Text | Diagnosis Prediction | | Examination Recommendation | | Treatment Recommendation | |
|------|--------|:---:|:---:|:---:|:---:|:---:|:---:|:---:|
| | | | Micro | Macro | Micro | Macro | Micro | Macro |
| Non-LLM | LSTM | ✓ | 74.0(2.0) | 73.1(2.0) | 75.7(0.3) | 76.4(0.2) | 56.7(0.6) | 50.0(0.7) |
| | mtGAN | ✓ | 81.9(2.2) | 80.9(2.5) | 72.4(1.5) | 73.4(1.4) | 58.6(2.8) | 52.9(3.0) |
| LLM-Based | MedSyn | ✓ | 81.7(0.0) | 81.7(0.0) | 82.9(0.1) | 82.2(0.1) | 74.5(0.1) | 71.3(0.2) |
| | LLM Direct | ✗ | 81.8(0.0) | 81.8(0.2) | 64.4(0.0) | 65.4(0.0) | 60.9(0.2) | 59.0(0.3) |
| | Self-Refine | ✗ | 81.9(0.3) | 81.8(0.3) | 64.9(0.8) | 65.7(0.7) | 63.1(0.3) | 61.3(0.5) |
| | **LLM-CARe(ours)** | ✗ | **82.6**(0.3) | **82.4**(0.4) | **85.3**(0.1) | **85.2**(0.1) | **76.9**(0.3) | **74.1**(0.5) |

To evaluate the utility of synthetic EMRs, we assess their effectiveness in training models for downstream tasks. As shown in Table 2, **LLM-CARe achieves the best performance across all three tasks, without training or prompting on any real EMR text**. In contrast, most baselines rely on

real records for model training or in-context examples, which may raise privacy concerns (privacy evaluation is shown in Appendix I). While downstream tasks do not map directly to quality, the overall trend is consistent: methods with higher quality also tend to perform better on downstream tasks. For example, MedSyn ranks second to ours on both quality and tasks. Notably, baseline methods perform relatively well on diagnosis but show larger gaps on examination and treatment tasks. We attribute this to their higher coverage of symptom-related knowledge but limited representation of clinical concepts related to examinations, procedures, and medications—key to the latter two tasks. These results highlight the advantage of our structured framework in producing semantically rich and clinically useful synthetic records. Further experiments using an alternative backbone and an external dataset are provided in Appendix J, with subgroup analyses shown in Figure 13.

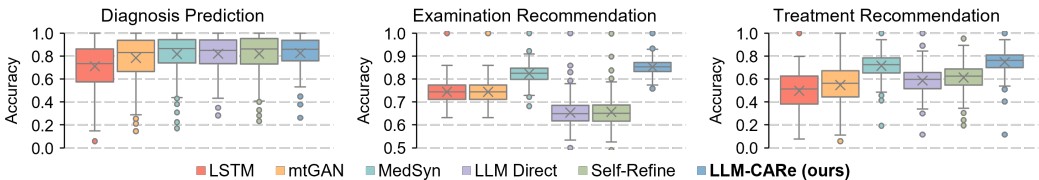

Figure 6: Accuracy distribution across diseases of different methods on three downstream tasks.

Figure 6 further illustrates the performance distribution across diseases. The box plots show that our method not only achieves higher average performance but also exhibits narrower variance across diseases. This consistency suggests that **our approach is broadly effective and robust across a wide range of clinical conditions**. In contrast, some baselines display wide performance fluctuations, indicating limited generalization to diverse disease types. These findings underscore the reliability of our method in real-world clinical settings, where robustness across varied diseases is critical.

### 5.3 Analysis of Performance Across Stages

Figure 7 shows how EMR quality and downstream task performance evolve through the three refinement stages. Quality dimensions improve most notably in their corresponding stages (e.g., completeness during the section stage). While some dimensions may show temporary fluctuations at other stages, the staged design—progressing from softer corpus-level constraints to stricter document-level checks—ensures that all dimensions ultimately exceed direct generation by a clear margin. For downstream tasks, examination and treatment recommendation benefit most from corpus-level refinement, since they rely on broad and diverse clinical concepts present in the training data. In contrast, diagnosis prediction depends more directly on complete histories and symptom–diagnosis alignment, thus improves primarily at the section and document stages.

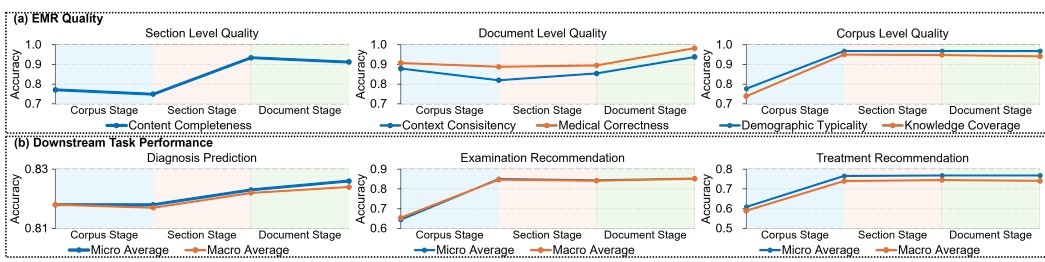

Figure 7: Trends on **(a)** EMR quality and **(b)** downstream task performance across stages.

### 5.4 Ablation Study of Multi-Agent Components

To assess the contribution of each agent in our framework, we conduct an ablation study by individually removing the Critic, Adviser, and Reviser agents. When the Critic is removed, the Adviser generates feedback for all quality criteria, regardless of whether they are already satisfied. Without

the Adviser, the Reviser receives only high-level information about unmet criteria, without actionable suggestions. When the Reviser is removed, the system cannot update existing drafts—instead, we prompt the Generator to regenerate EMRs using all quality criteria as input.

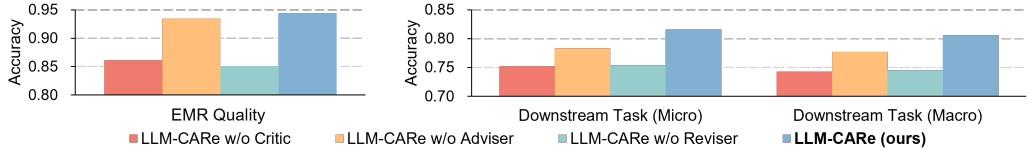

Figure 8: Impact of removing each agent on EMR quality and downstream task performance.

Figure 8 shows that removing any of the agents leads to a noticeable performance drop in both EMR quality and downstream tasks. The most significant declines occur when either the Critic or Reviser is ablated, highlighting two key insights: accurate assessment of the current draft is crucial for targeted refinement; and **large language models struggle to satisfy all quality criteria in a single generation step**, underscoring the need for cyclic refinement. Besides, removing the Adviser also results in a performance drop, suggesting that concrete, actionable feedback is more effective than abstract criterion-level input in guiding successful revisions.

## 5.5 ANALYSIS OF ROBUSTNESS ACROSS LLM BACKBONE

Table 3 compares our framework with the LLM Direct baseline across three different LLM backbones: a general-purpose model (LLaMA 3.1) (Dubey et al., 2024), a medically pre-trained model (Meditron) (Chen et al., 2023), and a reasoning-oriented model (R1) (Guo et al., 2025). Without any prompt tuning or model-specific adaptation, our method consistently improves both EMR quality and downstream task performance across all backbones.

Notably, **although Meditron is explicitly trained for medical domains, it still struggles to directly generate high-quality EMRs** and gains substantial improvements when integrated into our framework. Similarly, R1 does not significantly outperform the general model in direct generation, indicating that **internal reasoning alone is insufficient to meet the nuanced requirements of EMR**. These findings emphasize the necessity of principle-driven refinement that complements backbone capabilities and cannot be replaced by pretraining or reasoning alone.

Table 3: EMR quality and downstream performance (%) across LLM backbones, averaged over principles and tasks.

| Backbone | Generation Strategy | EMR Quality | Downstream Task Micro Average | Macro Average |
|---|---|---|---|---|
| Llama3.1 -8B-Instruct | LLM Direct | 49.3 | 54.9 | 55.8 |
| | **LLM-CARe** | **77.5** | **73.1** | **71.7** |
| Meditron3 -8B | LLM Direct | 53.9 | 53.7 | 54.8 |
| | **LLM-CARe** | **76.4** | **73.6** | **72.4** |
| R1-Distill -Llama-8B | LLM Direct | 55.5 | 51.3 | 52.4 |
| | **LLM-CARe** | **80.5** | **72.8** | **71.5** |

## 5.6 EFFECT ON INCORPORATING REAL EMR TEXT

To further examine whether LLM-CARe benefits from access to real EMR text, we introduce a variant in which the initial draft generator is given a real EMR as a reference example. As illustrated in Figure 9, the performance of this variant remain highly similar to the original LLM-CARe. These findings show that the effectiveness of LLM-CARe arises from its structured multi-agent cyclic refinement, which leads to strong performance without relying on real EMR text. Additional comparisons with other variants are provided in Appendix K.

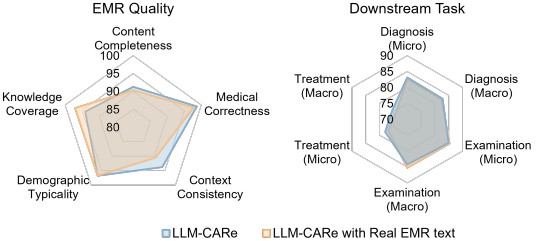

Figure 9: Comparison between LLM-CARe and a variant that incorporates real EMR text as an example during initial draft generation.

## 5.7 CLINICIAN EVALUATION

To validate the reliability of using LLM as a judge, we conducted a human evaluation study. A total of 200 synthetic EMRs were sampled (40 from each of five methods) and independently assessed by four licensed clinicians, who rated completeness, consistency, and correctness for each record. As shown in Table 4, the agreement

Table 4: Agreement between human clinicians and LLM-based evaluation on EMR quality.

| Quality Level | Clinician-LLM Agreement | | Inter-Clinician Agreement | |
|---|---|---|---|---|
| | Cohen's Kappa | Confidence Interval (95%) | Fleiss's Kappa | Confidence Interval (95%) |
| Section | 0.866 | [0.834, 0.896] | 0.975 | [0.959, 0.988] |
| Document | 0.797 | [0.764, 0.829] | 0.932 | [0.912, 0.950] |
| Overall | 0.833 | [0.809, 0.855] | 0.950 | [0.937, 0.962] |

between clinicians and LLM is consistently high (Cohen's Kappa = 0.833 overall, where values exceeding 0.8 indicate near-perfect agreement), with tight confidence intervals. Inter-clinician agreement is also strong (Fleiss's Kappa = 0.950 overall), confirming that the evaluation criteria are well-defined and consistently interpretable by human experts. Together, these results demonstrate that the **LLM-based evaluation closely aligns with human judgment, supporting its validity as an efficient proxy for large-scale quality assessment**. To ensure the stability of human evaluation, we gradually expanded the annotated subset from 100 to 200 EMRs and observed consistently stable agreement levels. Detailed results are reported in Appendix L.

## 5.8 CASE STUDY

Table 5 presents examples of quality issues that commonly arise when generation methods lack explicit adherence to quality standards. These cases reveal that without structured quality control, generated EMRs often exhibit missing details, medical inaccuracies, or inconsistencies.

In contrast, Figure 10 demonstrates how LLM-CARe progressively improves draft quality through refinement on different levels. This underscores the importance of stage-wise cyclic refinement in producing high-quality EMRs.

Table 5: Examples of quality issues in synthetic EMRs. Abbreviations: CC-Chief Complaint, HPI-History of Present Illness, HC-Hospital Course, Dx-Diagnosis.

| Method | Example | Problem |
|---|---|---|
| LSTM | **Dx**: Uterine leiomyomas **Gender**: Male | Males do not have a uterus. |
| mtGAN | **HC**: Discharged after feeling stable. | No treatments are mentioned in HC. |
| MedSyn | **CC**: Diarrhea for 2 days. **HPI**: ... no diarrhea ... | CC mentions diarrhea, but HPI denies it. |

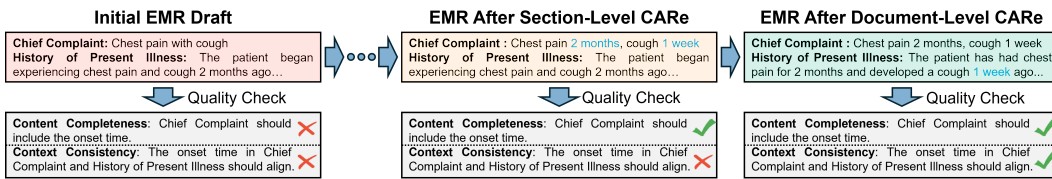

Figure 10: Illustration of quality improvements through LLM-CARe. Revisions are marked in blue.

## 6 CONCLUSION

In this work, we tackle the limitations of existing EMR synthesis methods which mimic real records without considering quality requirements. To overcome these, we propose **LLM-CARe**, a stage-wise cyclic refinement framework driven by the collaboration of **Critic**, **Adviser**, and **Reviser** agents. Instead of single-pass generation, LLM-CARe progressively enhances drafts through three dedicated stages: aligning corpus-level distributions, ensuring section-level completeness, and enforcing document-level consistency and correctness. Experiments on a large real-world dataset demonstrate that LLM-CARe substantially improves the quality of EMRs across all granularities. Moreover, models trained on the refined synthetic corpus achieve superior performance on various downstream tasks, highlighting the practical value of our approach. These results show the effectiveness of LLM-CARe in generating synthetic EMRs that are both high-quality and clinically meaningful, offering a reliable and privacy-preserving foundation for healthcare AI development.

## ACKNOWLEDGEMENT

This research was supported by Beijing Natural Science Foundation (NO. 4252046) and Non-communicable Chronic Diseases-National Science and Technology Major Project (Grant No. 2023ZD0506501).

## ETHICS STATEMENT

This work adheres to the ICLR Code of Ethics. This work focuses on improving the quality of synthetic EMRs guided by clinical quality principles. The core methodology does not involve training on actual EMRs. During evaluation, a limited set of test cases was accessed within a secure, institutional data environment. These records had been fully de-identified by the hosting healthcare organization and remained within its controlled data management platform. The study did not entail any active data collection from patients or clinicians. All data usage adhered to institutional policies and was conducted under the oversight of the relevant data governance framework.

## REPRODUCIBILITY STATEMENT

The collection and preprocessing of the EMR dataset are described in Section 4 and Appendix B. Experimental settings, model configurations, and evaluation protocols are detailed in Section 4 and Appendix E. Our code is available at `https://github.com/THUMLP/LLM-CARe`.

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

# A    DETAILS OF EMR QUALITY PRINCIPLES

In this section, we provide a detailed description of each criterion corresponding to the EMR quality principles, along with representative examples.

## A.1    CONTENT COMPLETENESS

Table 6 lists the criteria used to assess content completeness, which evaluate whether each field contains all essential information. The criteria follow standard clinical documentation conventions and use ordinary clinical terms (e.g., "major symptoms" refers to the primary complaints driving the visit, as opposed to secondary associated manifestations).

Table 6: Description of quality criteria for content completeness, along with representative examples. Abbreviations: CC-Chief Complaint, HPI-History of Present Illness, HC-Hospital Course, DI-Discharge Instructions.

| Criterion | Abbreviation | Positive Example | Negative Example |
|---|---|---|---|
| Chief complaint states reason for admission | CC Reason | • CC: Cough for 1 day
• CC: Thyroid nodule noted for 2 months | • CC: Admitted on 2025/05/16 |
| Chief complaint includes onset time | CC Onset | • CC: Fever for 6 days
• CC: Chest pain for 1 year, worsened over past month | • CC: Dizziness accompanied by nausea
• CC: Poor recent glycemic control |
| History of present illness describes acuity of onset | HPI Acuity | • HPI: Sudden-onset headache 1 week ago
• HPI: Gradual onset of unsteady gait for 4 months | • HPI: Experienced headache over a month ago
• HPI: Developed gait instability recently |
| History of present illness mentions possible causes | HPI Cause | • HPI: Abdominal pain after alcohol intake 1 day ago
• HPI: Dizziness for 2 weeks without obvious cause | • HPI: Sudden right eye vision loss one week ago
• HPI: Cough onset 3 days ago |
| History of present illness lists major symptoms and onset time | HPI Symptom | • HPI: Vomited 4–5 times over the past half day
• HPI: Poor appetite and fatigue over past 2 weeks | • HPI: Experienced dizziness for days |
| History of present illness includes all general conditions | HPI General | • HPI: Normal mental status, sleep, appetite, bowel and bladder function; no significant weight change | • HPI: Appetite decreased |
| Hospital course includes auxiliary examinations or laboratory examinations | HC Examination | • HC: Chest CT revealed a pulmonary mass lesion
• HC: Admission labs showed CRP: 12.3 mg/L | • HC: Patient underwent further examinations after admission |
| Hospital course includes treatment interventions | HC Treatment | • HC: Appendectomy under general anesthesia
• HC: Aspirin given for antiplatelet therapy | • HC: Given pharmacological therapy |
| Discharge instruction includes medication dosage and usage | DI Medication | • DI: Atorvastatin 1 tablet nightly
• DI: Amoxicillin 1g twice daily | • DI: Take antibiotics regularly |

## A.2    MEDICAL CORRECTNESS

Table 7 outlines the criteria for medical correctness, which assess whether the clinical content aligns with the patient's diagnosis.

Table 7: Description of quality criteria for medical correctness, along with representative examples. Abbreviations: CC-Chief Complaint, HPI-History of Present Illness, HC-Hospital Course, DI-Discharge Instructions, Dx-Diagnosis, PD-Patient Demographics.

| Criterion | Abbreviation | Positive Example | Negative Example |
|---|---|---|---|
| Diagnosis matches the patient's gender | Dx-PD Gender | • Dx: Pregnancy Gender: Female | • Dx: Pregnancy Gender: Male |
| Symptoms in chief complaint align with diagnosis | Dx-CC Symptom | • Dx: Pneumonia CC: Cough for 1 day | • Dx: Pneumonia CC: Knee pain for 3 days |
| Symptoms in history of present illness align with diagnosis | Dx-HPI Symptom | • Dx: Cerebral infarction HPI: Sudden slurred speech 1 day ago | • Dx: Acute appendicitis HPI: Sudden blurred vision 2 weeks ago |
| Examinations in hospital course align with diagnosis | Dx-HC Examination | • Dx: Pneumonia HC: Chest CT indicated pneumonia | • Dx: Cerebral infarction HC: Abdominal ultrasound showed appendiceal thickening |
| Medications in discharge instructions align with diagnosis | Dx-DI Medication | • Dx: Type 2 diabetes DI: Metformin (0.5g), one tablet twice daily | • Dx: Pneumonia DI: Insulin injection before meals |

## A.3 CONTEXT CONSISTENCY

Table 8 presents the criteria for context consistency, which evaluate whether information across different EMR sections is logically coherent.

Table 8: Description of quality criteria for context consistency, along with representative examples. Abbreviations: CC-Chief Complaint, HPI-History of Present Illness, HC-Hospital Course.

| Criterion | Abbreviation | Positive Example | Negative Example |
|---|---|---|---|
| Symptoms in chief complaint are consistent with those in history of present illness | CC-HPI Symptom | • CC: Cough for 1 day HPI: ... wich cough ... | • CC: Cough for 1 day HPI: ... wichout cough ... |
| Onset time in chief complaint is consistent with that in history of present illness | CC-HPI Onset | • CC: Chest pain for 1 month HPI: Chest pain over past 1 month | • CC: Chest pain for 1 month HPI: Chest pain over past 2 months |
| Affected site in history of present illness is consistent with the site of examination or treatment in hospital course | HPI-HC Site | • HPI: Left leg pain after a fall HC: X-ray showed a fracture of the left leg. | • HPI: Left leg pain after a fall HC: X-ray showed a fracture of the right leg. |

## A.4 DEMOGRAPHIC TYPICALITY

For demographic typicality, we focus on two key patient attributes: gender and age. We evaluate whether the distributions of these attributes in the synthetic EMRs align with the target distributions. Figure 11 illustrates representative examples of gender and age distributions that are aligned with and deviate from the target distribution.

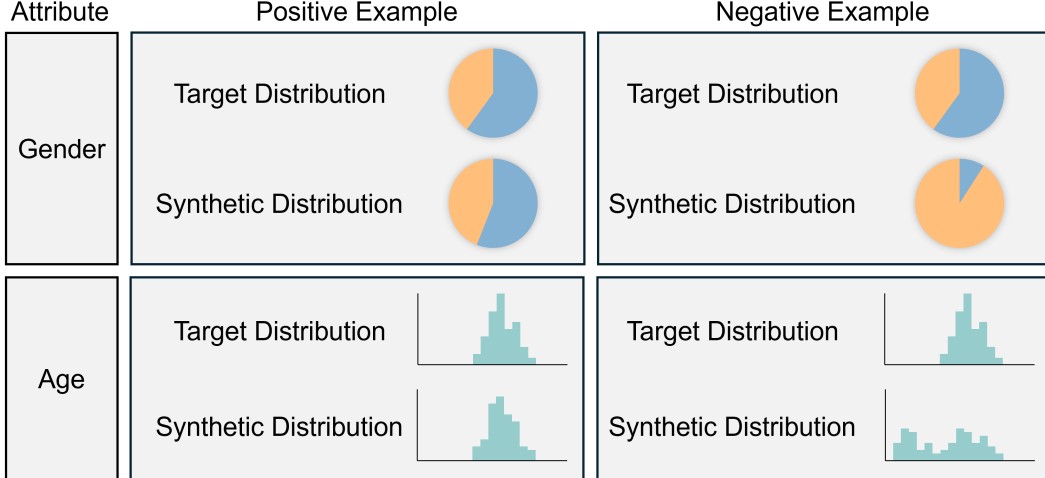

Figure 11: Examples of synthetic data distributions that are either consistent or inconsistent with the target distribution.

### A.5 KNOWLEDGE COVERAGE

For knowledge coverage, we focus on five key categories of clinical knowledge: symptoms, auxiliary examinations, laboratory examinations, surgeries, and medications. Table 9 lists representative entities from each category.

Table 9: Knowledge categories and representative entities.

| Category | Example |
|---|---|
| Symptom | Cough, Fever, Headache, Nausea, ... |
| Auxiliary Examination | Chest CT, Brain MRI, ECG, Abdominal Ultrasound, ... |
| Laboratory Examination | Complete Blood Count, Liver Function Test, C-Reactive Protein, ... |
| Surgery | Appendectomy, Tonsillectomy, Cataract Surgery, Cholecystectomy, ... |
| Medication | Aspirin, Penicillin, Metformin, Atorvastatin, ... |

## B DETAILS OF DATASET

### B.1 DATASET CONSTRUCTION

We conduct our experiments on a large-scale real-world EMR dataset containing 1.82 million de-identified medical records collected from hospitals. Personally identifiable information (e.g., patient and clinician names, phone numbers, locations) had already been removed by the data provider, ensuring compliance with privacy standards. All experiments were conducted on hospital-controlled infrastructure to ensure data security and prevent risk of privacy leakage.

To ensure data quality, we first remove records that are missing critical information, such as patient age or gender, primary diagnosis, or any of the four target fields: chief complaint, history of present illness, hospital course, and discharge instruction. After this filtering step, 905k records remain.

We then apply a length-based filtering criterion to further improve data quality. During inspection, we found that overly short entries often contain placeholders or incomplete content, while excessively long entries are more likely to include unintelligible text. Therefore, we retain only records where the chief complaint is under 20 words, and each of the other fields falls within the 10 to 1,000 word range. This step yields a subset of 710k high-quality records.

Lastly, to ensure the reliability and stability of downstream evaluation, we retain only common diseases with sufficient data volume. Specifically, we exclude any diagnosis category with fewer than 500 records. This ensures that each disease has at least 100 samples in the test set after an 80/20 train-test split. Diseases with very few examples can lead to high-variance estimates and lack statistical significance in evaluation. After this final filtering step, we obtain 192k EMRs spanning 302 distinct disease categories.

### B.2 CONSTRUCTION OF DOWNSTREAM TASKS

For all downstream tasks, the correct options are extracted directly from the EMR. For the diagnosis prediction task, incorrect options are randomly sampled from other diagnoses in the dataset. For the test and treatment prediction tasks, incorrect options are selected to be incompatible with the gold diagnosis: we first exclude all tests or treatments that appear in EMRs with the same diagnosis, and then randomly sample from the remaining pool. Using this approach, we construct training examples from synthetic EMRs, and evaluate model performance on questions built from real EMRs in the held-out test set. The diagnosis prediction task contains exactly one question per EMR, yielding 38k samples in the test set. For examination and treatment recommendation, each extracted examination or treatment entity forms one question, resulting in 346k and 110k test questions respectively.

## C BASELINE IMPLEMENTATION DETAILS

For the non-LLM baselines (LSTM and mtGAN), we trained both models from scratch on the training set. Each model receives a one-hot disease vector as input and is optimized to generate the corresponding real EMR, treated as the supervision target.

For the LLM-based baselines (MedSyn, LLM Direct, and Self-Refine), we used the same backbone model as in LLM-CARe (Qwen2.5-7B-Instruct) to ensure a fair comparison. **MedSyn**: For each generation, we randomly sample one real EMR of the target disease from the training set as an in-context example, and randomly sample five symptoms associated with the disease as additional cues. The model then generates an EMR in a single pass conditioned on these inputs. **LLM Direct**: We generate EMRs using the same initial-draft prompt template as in LLM-CARe, without applying any refinement or evaluation steps. **Self-Refine**: We aggregate all section-level and document-level criteria into a single combined instruction. The model first identifies issues based on this unified prompt and then produces a revised EMR accordingly, without multi-level decomposition.

## D EVALUATION DETAILS

For the LLM-based evaluation of EMR quality, we prompt the model to assess each generated EMR against the predefined criteria for medical correctness, content completeness, and context consistency. Each criterion is formulated as a binary classification task—whether a given EMR satisfies the criterion or not. The model outputs a yes/no response for each criterion per EMR, and we compute the final score by averaging over all EMRs.

For demographic typicality, we compare the distribution of demographic attributes in synthetic EMRs to those in the real dataset. For gender, we use the total variation distance (TVD) between the two distributions. For age, which is a continuous variable, we compute the Wasserstein distance. Since lower distance values indicate higher similarity, we transform the scores by computing $1 - \text{TVD}$ and $1 - \text{Wasserstein}$, respectively, so that higher values consistently reflect better quality across all metrics.

For knowledge coverage, we first extract medical entities associated with each diagnosis from the real EMRs. We then measure the proportion of these entities that appear in synthetic EMRs with the same diagnosis. To avoid the complexity and potential noise of semantic matching, we use exact string-level matching to compare entity presence.

For the downstream task evaluation, we report both macro and micro accuracy. Macro accuracy averages the model performance across all diagnoses by first computing the accuracy within each disease category, then averaging across categories. Micro accuracy, in contrast, computes the overall

accuracy across all samples regardless of diagnosis. This dual evaluation provides a comprehensive view of model generalizability across frequent and less frequent disease types.

# E EXPERIMENTAL DETAILS

## E.1 PROMPTS USED FOR EACH AGENT

In this section, we list the prompts used for each agent in our LLM-CARe framework.

**Generator**

> Please generate an electronic medical record according to the following requirements:
> 1. The patient's primary diagnosis is: [diag].
> 2. Include only the following sections: 'Gender', 'Age', 'Primary Diagnosis', 'Chief Complaint', 'History of Present Illness', 'Hospital Course', and 'Discharge Instructions'.
> 3. Section-specific instructions:
> - The Chief Complaint should briefly describe the reason for admission.
> - The History of Present Illness should describe the onset and development of the condition in detail.
> - The Hospital Course should mention the examinations and treatments the patient received.
> - The Discharge Instructions should specify post-discharge recommendations, such as prescribed medications.
> 4. Output the result in JSON format with the structure: "Section Name": "Section Content", where each section content is a single string.

**Section-Level Critic**

> Below is the '[section_name]' section from an electronic medical record:
> [section]
> Please determine whether the above '[section_name]' meets the following requirement: [requirement].
> Respond in the following JSON format:
> {"Meets Requirement": true/false}

**Section-Level Adviser**

> Below is the '[section_name]' section from an electronic medical record:
> [section]
> This section does not meet the following requirement: [requirement]. Please provide a specific revision suggestion based on the section content, explaining how it should be modified to meet the requirement.
> Respond in the following JSON format:
> {"Revision Suggestion": "specific suggestion"}

**Section-Level Reviser**

> Below is the '[section_name]' section from an electronic medical record with an issue:
> [section]
> The '[section_name]' section misses essential content. Please revise the record based on the following suggestion: [feedback]
> Return the result in JSON format using the pattern "Section Name": "Section Content", and include only the "[section_name]" section. The content of the section should be a single string.

**Document-Level Critic**

Below is an electronic medical record consisting of multiple sections:

[record]

Please evaluate whether the record satisfies the following requirement: [requirement]. Identify any conflicts or implausible statements across sections.

Respond in the following JSON format:

{"Meets Requirement": true/false}

**Document-Level Adviser**

The following record has issues violating the requirement: [requirement]:

[record]

Please provide targeted suggestions to resolve the problem. Prioritize changes to the sections that minimally disrupt overall coherence.

Respond in the following JSON format:

{"Revision Suggestion": "specific suggestion"}

**Document-Level Reviser**

Below is an electronic medical record with flagged issues:

[record]

Please revise the record according to the following suggestion: [feedback].

Return the updated record in JSON format, including all sections. The content of each section should be a single string.

**Corpus-Level Agents**: For the corpus-level stage, we use statistical analysis tools as the critic and adviser rather than LLMs. Therefore, no natural language instructions are required for these agents; their operations are fully automated and operate on dataset-wide distributions. For the corpus-level reviser, each sample in the selected subset is modified individually, using the same type of instructions as the document-level reviser.

### E.2 LLM Backbones

We use the following pretrained large language models in our experiments:

- **Qwen2.5** (Yang et al., 2025): Licensed under the Apache 2.0 License[1]. We use the model checkpoints available on Huggingface[2].

- **LLaMA 3.1** (Dubey et al., 2024): Licensed under the LLaMA 3.1 Community License[3]. We use the model checkpoints available on Huggingface[4].

- **Meditron 3** (Chen et al., 2023): Licensed under the LLaMA 3.1 Community License[5]. We use the model checkpoints available on Huggingface[6].

- **DeepSeek-R1-Distill-Llama** (Guo et al., 2025): Licensed under the MIT License[7]. We use the model checkpoints available on Huggingface[8].

---

[1]https://huggingface.co/Qwen/Qwen2.5-7B-Instruct/blob/main/LICENSE
[2]https://huggingface.co/Qwen
[3]https://www.llama.com/llama3_1/license/
[4]https://huggingface.co/meta-llama/Llama-3.1-8B-Instruct
[5]https://www.llama.com/llama3_1/license/
[6]https://huggingface.co/OpenMeditron/Meditron3-8B
[7]https://github.com/deepseek-ai/DeepSeek-R1/blob/main/LICENSE
[8]https://huggingface.co/deepseek-ai/DeepSeek-R1-Distill-Llama-8B

### E.3 HYPERPARAMETERS

For LLM-CARe and LLM Direct—we adopt the default generation configuration provided with each model checkpoint. For all refinement stages (corpus, section, and document), the Critic, Adviser, and Reviser agents are iterated for two cycles before proceeding to the next stage. We empirically observed that additional iterations beyond two provided negligible improvements in EMR quality and downstream task performance.

For EMR quality evaluation, we use greedy decoding to ensure deterministic outputs. For downstream tasks, we fine-tune Qwen2.5-0.5B-Instruct using the AdamW optimizer with a batch size of 16, a learning rate of 2e-5, and a cosine learning rate scheduler with 5% warmup steps. The model is fine-tuned for 3 epochs on the synthetic dataset and evaluated on the real test set.

### E.4 SOFTWARE

EMR generation with LLMs is conducted using vLLM (Kwon et al., 2023). PyTorch (Paszke et al., 2019) is used for training and inference of non-LLM baselines. Fine-tuning of downstream task models is performed using the Huggingface Transformers library (Wolf et al., 2020). Evaluations are also executed using vLLM.

### E.5 COMPUTATIONAL RESOURCES

All experiments—except for EMR quality evaluation—are conducted on NVIDIA RTX 4090 GPU with 24GB of memory. EMR quality evaluation, which uses Qwen2.5-32B-Instruct, is performed on NVIDIA A100 GPU with 80GB of memory.

## F ROBUSTNESS OF LLM-AS-JUDGE EVALUATION

Table 10: Quality scores evaluated by different LLMs.

| Evaluation LLM | Method | Content Completeness | Medical Correctness | Context Consistency |
|---|---|---|---|---|
| Qwen2.5-32B | LSTM | 70.8 | 65.0 | 21.7 |
| | mtGAN | 55.8 | 51.8 | 21.4 |
| | MedSyn | 84.8 | 95.3 | 91.9 |
| | LLM Direct | 77.1 | 90.7 | 87.9 |
| | LLM-CARe (ours) | **91.2** | **98.6** | **93.8** |
| GPT-OSS-20B | LSTM | 73.9 | 88.9 | 8.9 |
| | mtGAN | 62.8 | 87.7 | 13.4 |
| | MedSyn | 79.9 | 97.3 | 86.9 |
| | LLM Direct | 81.1 | 96.7 | 70.2 |
| | LLM-CARe (ours) | **93.4** | **99.0** | **91.3** |
| Deepseek-V3.2-Exp | LSTM | 69.9 | 72.5 | 2.6 |
| | mtGAN | 57.9 | 64.0 | 0.7 |
| | MedSyn | 80.6 | 97.5 | 85.0 |
| | LLM Direct | 79.9 | 96.8 | 77.8 |
| | LLM-CARe (ours) | **91.1** | **98.5** | **89.8** |

To examine whether our intrinsic quality evaluation is robust to the choice of LLM judge, we performed an extended cross-model assessment using three independent evaluators: Qwen2.5-32B, GPT-OSS-20B (OpenAI, 2025), and DeepSeek-V3.2-Exp (DeepSeek-AI, 2025). Qwen2.5-32B is our original judge; GPT-OSS-20B was included due to its reported strength in medical-language understanding tasks, and DeepSeek-V3.2-Exp (API) was added as a large commercial model to further test evaluation generality across architectures and training regimes. Owing to the computational cost of the API, DeepSeek evaluation was conducted on 10% sampled data.

All three evaluators were provided with the same scoring instructions used in our main experiments. As shown in Table 10, the absolute scores vary across models—as expected due to differences in calibration and alignment—but the overall ranking pattern remains consistent, with LLM-CARe achieving the top scores across all intrinsic quality dimensions under all judges. The baselines also maintain broadly similar relative positions, with only minor ordering differences between closely performing systems. These results indicate that our evaluation conclusions are not tied to a specific model family and that LLM-CARe demonstrates robust superiority under multiple independent evaluators.

## G  SUBGROUP ANALYSIS ACROSS DEMOGRAPHIC AND DISEASE FREQUENCY

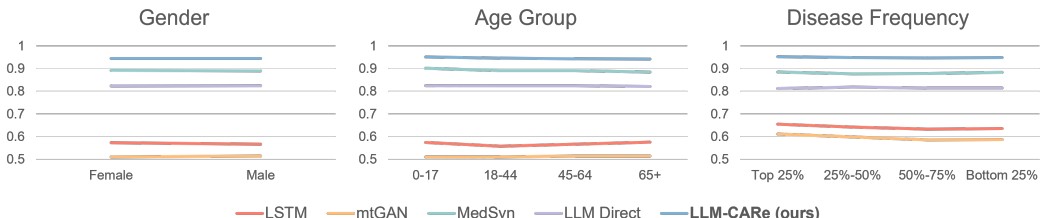

Figure 12: Subgroup analysis of intrinsic EMR quality.

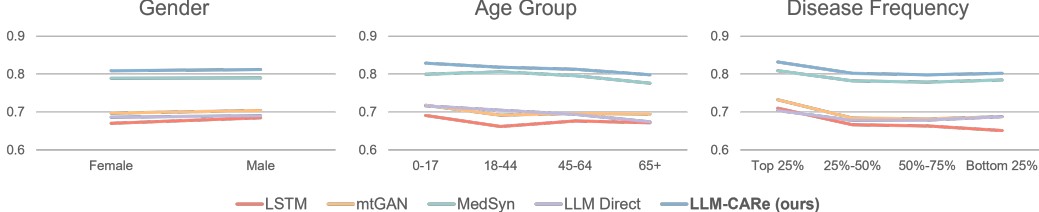

Figure 13: Subgroup analysis of downstream task performance.

To examine whether LLM-CARe amplifies or mitigates subgroup disparities, we conducted a stratified evaluation across three grouping dimensions: gender, age (0–17, 18–44, 45–64, 65+), and disease-frequency strata. Figure 12 reports intrinsic EMR quality across subgroups, and Figure 13 shows downstream performance under the same partitions. LLM-CARe consistently achieves the highest scores across all subgroups, and the variation across demographic and frequency groups is comparable to or smaller than that of the baselines. These results indicate that LLM-CARe does not reinforce demographic or clinical biases and maintains robustness across diverse subpopulations, supporting its applicability to balanced synthetic EMR corpus construction.

## H  FULL RESULTS OF EMR QUALITY

Figure 14 provides the complete breakdown of EMR quality across all evaluated criteria, extending the representative results presented in Figure 5 of the main text. These detailed results offer a more comprehensive view of how different methods perform with respect to each quality dimension.

## I  PRIVACY EVALUATION VIA MEMBERSHIP INFERENCE ATTACK

To assess whether synthetic EMRs generated by different methods inadvertently reveal information from real patient records, we conduct a membership inference attack (MIA)—a standard privacy evaluation technique that tests whether an attacker can distinguish training examples from non-

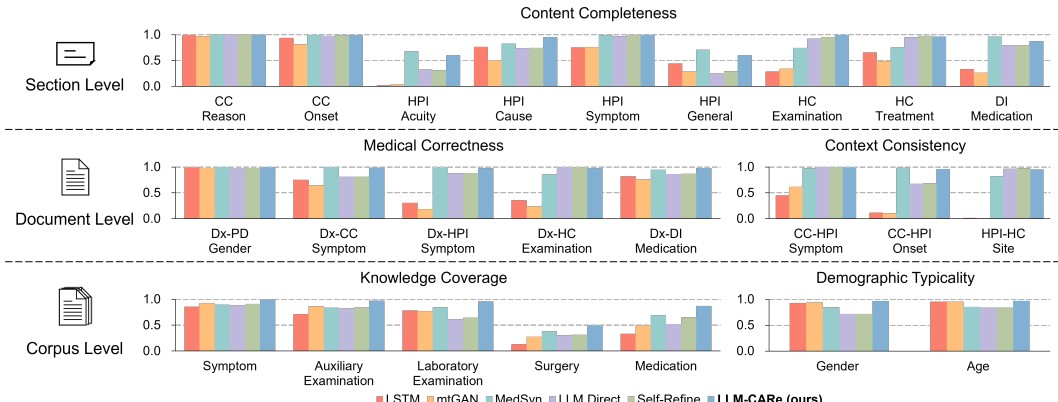

Figure 14: Fine-Grained EMR quality evaluation across three level of criteria. Abbreviations of EMR sections: CC-Chief Complaint, HPI-History of Present Illness, HC-Hospital Course, DI-Discharge Instructions, Dx-Diagnosis.

members. An accuracy of 0.50 corresponds to random guessing; values close to 0.50 therefore indicate stronger privacy, as the attacker cannot reliably infer membership.

Table 11 reports the attack accuracy across all methods. LLM-CARe and LLM Direct both achieve values near 0.50, as expected, since neither method accesses real EMR text during generation. LSTM and mtGAN also remain close to 0.50. Although these models are trained on real EMRs, their limited model capacity—combined with the complexity of multi-section EMRs used in our setting—reduces their ability to memorize full clinical notes. In contrast, MedSyn, which directly uses real EMRs as in-context exemplars, exhibits higher attack accuracy, indicating elevated privacy risk due to closer exposure to specific real records.

Table 11: Membership inference attack accuracy (closer to 0.50 is better).

| Method | Attack Accuracy |
|---|---|
| LSTM | 0.499 |
| mtGAN | 0.504 |
| MedSyn | 0.533 |
| LLM Direct | 0.500 |
| LLM-CARe (ours) | 0.504 |

These results show that LLM-CARe introduces no observable privacy risk and that its generation process remains indistinguishable from non-member data, consistent with the fact that it operates without using any real EMR text.

## J ADDITIONAL VALIDATION OF DOWNSTREAM TASK RELIABILITY

To further examine the robustness of downstream task results, we conducted two complementary evaluations.

Table 12: Downstream task performance using Llama-3.2-1B-Instruct as the backbone model.

| Method | Diagnosis Prediction | | Examination Recommendation | | Treatment Recommendation | |
|---|---|---|---|---|---|---|
| | Micro | Macro | Micro | Macro | Micro | Macro |
| LSTM | 68.5 | 67.1 | 74.7 | 75.4 | 55.6 | 48.6 |
| mtGAN | 78.8 | 77.7 | 72.8 | 73.6 | 58.9 | 53.1 |
| MedSyn | 77.2 | 77.8 | 82.4 | 81.7 | 72.0 | 68.7 |
| LLM Direct | 78.9 | 79.0 | 61.9 | 62.8 | 59.4 | 57.0 |
| **LLM-CARe (ours)** | **80.1** | **79.8** | **83.3** | **83.2** | **74.7** | **71.8** |

First, we replaced the backbone model used for downstream classifiers and retrained all methods using Llama-3.2-1B-Instruct. As reported in Table 12, the relative ordering of all methods remained

stable, with LLM-CARe achieving the highest accuracy on diagnosis prediction, examination recommendation, and treatment recommendation. This consistency under a different model architecture indicates that performance gains do not depend on properties of the Qwen family, but instead arise from the proposed multi-level refinement framework.

Table 13: Diagnosis prediction accuracy on MIMIC-IV-Note dataset.

| Method | Micro Average | Macro Average |
|---|---|---|
| LSTM | 0.587 | 0.552 |
| mtGAN | 0.589 | 0.555 |
| MedSyn | 0.584 | 0.571 |
| LLM Direct | 0.589 | 0.532 |
| **LLM-CARe (ours)** | **0.606** | **0.590** |

Second, to test whether models benefit from superficial alignment with the textual format of the generation prompts, we constructed an additional diagnosis-prediction task using MIMIC-IV-Note. We selected only records corresponding to the same disease categories but originating from a different clinical institution and written in a distinctly different narrative style. All models trained on synthetic corpora exhibit lower accuracy due to the distributional and stylistic shift, yet LLM-CARe again achieves the best performance (Table 13), demonstrating that the downstream advantages are not tied to prompt-format similarity but to improved clinical quality in the synthesized EMRs.

## K  ROBUSTNESS TO PROMPTING STYLES AND INITIAL DRAFT QUALITY

Table 14: Intrinsic EMR quality under prompt rephrasing and improved initial drafts.

| Method | Section Level | Document Level | | Corpus Level | |
|---|---|---|---|---|---|
| | Content Completeness | Medical Correctness | Context Consistency | Demographic Typicality | Knowledge Coverage |
| LLM-CARe (ours) | 91.2 | 98.6 | 93.8 | 96.8 | 94.1 |
| +Rephrased Prompts | 94.1 | 98.0 | 93.8 | 96.8 | 95.1 |
| +Better Initial Draft | 93.3 | 98.6 | 94.0 | 96.8 | 96.2 |

Table 15: Downstream prediction performance (micro/macro accuracy) under prompt and draft perturbations.

| Method | Diagnosis Prediction | | Examination Recommendation | | Treatment Recommendation | |
|---|---|---|---|---|---|---|
| | Micro | Macro | Micro | Macro | Micro | Macro |
| LLM-CARe (ours) | 82.6 | 82.4 | 85.3 | 85.2 | 76.9 | 74.1 |
| +Rephrased Prompts | 81.1 | 80.7 | 83.4 | 83.4 | 75.3 | 73.0 |
| +Better Initial Draft | 82.6 | 81.8 | 84.7 | 84.4 | 76.8 | 73.9 |

We evaluated the stability of LLM-CARe under variations in prompting style and initial draft quality. First, all agent prompts were substantially rephrased using GPT-5 while preserving only high-level intent. Second, to test sensitivity to draft quality, we replaced the LLM-Direct drafts with higher-quality MedSyn drafts. Tables 14 and 15 report intrinsic quality and downstream task performance.

Across both perturbations, numerical differences are small and the overall ranking of methods remains unchanged. This robustness reflects the structure of LLM-CARe: quality requirements are decomposed into explicit criteria, and each agent operates on a single criterion at a time, making the refinement cycle insensitive to prompt wording. Likewise, the staged refinement progressively corrects deficiencies from multiple dimensions, reducing dependence on the initial draft. These properties together ensure that moderate changes in prompt phrasing or draft quality do not materially affect the final refined EMRs.

## L  DETAILS OF CLINICIAN EVALUATION

### L.1  RELIABILITY OF CLINICIAN EVALUATION

Table 16: Agreement between clinicians and LLM-based evaluation across different sample sizes.

| # Samples | Clinician-LLM Agreement | | Inter-Clinician Agreement | |
|---|---|---|---|---|
| | Cohen's Kappa | Confidence Interval (95%) | Fleiss's Kappa | Confidence Interval (95%) |
| 100 | 0.837 | [0.804, 0.868] | 0.947 | [0.928, 0.964] |
| 150 | 0.842 | [0.816, 0.868] | 0.948 | [0.933, 0.962] |
| 200 | 0.833 | [0.809, 0.855] | 0.950 | [0.937, 0.962] |

To ensure that clinician evaluation reflects a broad range of clinical scenarios, all clinician-labeled samples were obtained through stratified sampling. Diseases were grouped by frequency, and each group contributed proportionally to the evaluation set. The sampling was likewise balanced across generation methods, gender, and age groups, ensuring that the annotated EMRs covered both common and less-common situations and represented diverse demographic and clinical patterns.

To examine the stability of human evaluation, we increased the evaluation size from 100 to 150 and then to 200 EMRs. As shown in Table 16, clinician–LLM agreement (Cohen's Kappa) remains consistently above 0.8 across all three subset sizes, and inter-clinician agreement (Fleiss's Kappa) stays above 0.9. Confidence intervals narrow as sample size increases, but the agreement values themselves remain highly similar, indicating that both human annotations and LLM-based evaluations are stable and reliable.

### L.2  EXAMPLES OF DISAGREEMENT BETWEEN LLM AND CLINICIANS

We present examples of disagreement observed in our human evaluation.

**LLM–Clinician Disagreement**
*Criterion*: Discharge instructions should specify both the medication name and its dosage/usage.
*Discharge Instruction*: Continue oral antibiotic therapy...
*LLM:* Correct
*Clinician:* Wrong

In such cases, the LLM treats non-specific statements (e.g., "continue oral antibiotic therapy") as sufficiently informative, whereas clinicians regard them as incomplete because essential details such as drug name and dosage are missing. This reflects the LLM's more permissive interpretation of medication-related content.

**Inter–Clinician Disagreement**
*Criterion:* The history of present illness should describe general conditions (diet, sleep, mental status, bowel/urination, weight change).
*History of Present Illness:* The patient developed dizziness and fatigue two days ago... Appetite and sleep are normal; bowel and urination regular; no weight change.
*Clinician A:* Correct
*Clinician B:* Wrong

Here, clinicians differ in how they interpret the requirement to document mental status. Some consider dizziness and fatigue to implicitly convey reduced alertness, whereas others expect an explicit statement. Such differences reflect variation in documentation style rather than clinical competence and represent a common source of mild disagreement in narrative EMR review.

### L.3 EVALUATION SCHEMA

To verify the reliability of our evaluation, we asked licensed clinicians to assess the quality of synthetic EMRs. Clinicians were instructed to read each synthetic record carefully and then answer several yes/no questions regarding **completeness**, **consistency**, and **correctness**. The questions were designed to be simple binary judgments to ensure reproducibility. The detailed labeling instructions are as follows:

> Please review the synthetic EMR text shown below.
>
> **Synthetic EMR:**
> **Gender:** Male
> **Age:** 45 years old
> **Primary Diagnosis:** Pneumonia
> **Chief Complaint:** Fever and cough for 3 days . . .
> **History of Present Illness:** Patient developed fever three days ago, accompanied by cough and mild chest pain . . .
> . . .
>
> Based on the above synthetic EMR, please answer the following questions. For each question, mark your judgment in the blank: `Yes` if the requirement is satisfied, `No` otherwise.
> **Completeness**
> - Does the history of present illness include major symptoms? Answer: ____ (`Yes/No`)
> - . . .
>
> **Consistency**
> - Are the symptoms in the chief complaint consistent with those in the history of present illness? Answer: ____ (`Yes/No`)
> - . . .
>
> **Correctness**
> - Is the patient's sex valid given the diagnosis? Answer: ____ (`Yes/No`)
> - . . .

**Confidence interval estimation:** To quantify agreement, we report Cohen's Kappa and Fleiss's Kappa with 95% confidence intervals. The intervals were computed using a non-parametric bootstrap procedure with 10,000 resamples, which provides uncertainty estimates without assuming normality of the statistics.

## M  COMPUTATIONAL COST

On a single RTX 4090 GPU, producing 38k synthetic EMRs takes approximately 13 hours with direct generation, whereas LLM-CARe requires about 36 hours, which includes initial draft generation and all three refinement stages. Although the framework performs three levels of Critic–Adviser–Reviser interactions, the actual overhead is moderated by two factors. First, the outputs of these agents are much shorter than full EMRs, making each refinement step relatively lightweight. Second, if a draft already satisfies the criteria at a given stage, it bypasses subsequent agents, avoiding unnecessary iterations. Moreover, the computational cost scales linearly with the number of EMRs, and the cyclic refinement process can be parallelized across disease categories, making the framework feasible for scaling to larger corpora.

## N  USE OF LLMS

In preparing this manuscript, we used LLM solely as an assistive tool for text refinement, including grammar correction, and language polishing. The research ideas, experimental design, implementation, and analysis were entirely conceived and executed by the authors.

