# OpenReview forum: "Critic–Adviser–Reviser Cyclic Refinement: Towards High-Quality EMR Corpus Generation with LLMs"
_ICLR.cc/2026/Conference — ICLR 2026 Poster_

### Official Review · Reviewer_3q4A · 2025-10-26

**Soundness:** 3
**Presentation:** 4
**Contribution:** 4
**Rating:** 8
**Confidence:** 4

**Summary:**

This paper proposes LLM-CARe (Critic–Adviser–Reviser Cyclic Refinement), a multi-agent, stage-wise framework for generating high-quality synthetic electronic medical records (EMRs) using large language models without access to real data. The framework operates at three granular levels—corpus, section, and document—each involving cyclic collaboration among three agents: a Critic that evaluates, an Adviser that provides feedback, and a Reviser that updates drafts. The stages sequentially enforce corpus-level distributional realism, section-level content completeness, and document-level logical and clinical consistency. Evaluations on a de-identified dataset of 192k EMRs across 302 disease categories demonstrate that LLM-CARe produces synthetic data of significantly higher intrinsic quality and downstream utility than GAN-based, autoregressive, and direct LLM baselines. Notably, synthetic corpora produced by LLM-CARe enable superior model performance on diagnosis, examination, and treatment prediction tasks, all without using any real EMRs for prompting or fine-tuning

**Strengths:**

- The paper articulates a pressing problem and approaches it with a structured, multi-agent refinement paradigm inspired by human editorial cycles. This Critic–Adviser–Reviser decomposition is conceptually elegant and operationally grounded.
- The division into corpus-, section-, and document-level quality control provides a principled way to address EMR quality at increasing semantic granularity. This explicit separation reflects a solid understanding of how clinical documentation structure affects utility and consistency.
- Extensive experiments compare LLM-CARe to autoregressive (LSTM), GAN-based (mtGAN), and LLM-based (MedSyn, LLM Direct) baselines. LLM-CARe consistently achieves top scores across all five quality dimensions (completeness, correctness, consistency, demographic typicality, and knowledge coverage) and outperforms on downstream tasks
- The framework improves EMR quality across several backbone models (LLaMA 3.1, Meditron, R1-Distill), showing that the refinement process generalizes beyond a single model or domain setting.
- The ablation study isolates each agent’s contribution, empirically confirming that cyclic interaction—not one-shot prompting—is essential to achieve quality improvements.

**Weaknesses:**

- While baselines like MedSyn and mtGAN rely on real EMRs for conditioning or training, LLM-CARe is evaluated without them, creating a methodological imbalance. A fair comparison should include a variant of LLM-CARe that also conditions on limited real data to isolate the contribution of cyclic refinement rather than data access differences.
- Please include a recent citation found related to LLM based synthetic EMR generation at a recent ML for health conference (https://proceedings.mlr.press/v287/lin25a.html)
- The paper asserts privacy preservation without providing empirical checks for memorization or data leakage—critical for synthetic data generation claims.
- Because the evaluation tasks (diagnosis, examination, treatment) are derived from the same structure as the generation prompts, there may be an inductive bias favoring models trained under similar textual formats. Try following a clinical task derived from MEDS or something unrelated (https://openreview.net/forum?id=IsHy2ebjIG)
- The multi-agent cyclic process is likely computationally heavy, but the paper does not report inference or generation latency, making it difficult to assess the feasibility of applying LLM-CARe to million-record corpora.
-

**Questions:**

- How do you control for potential overfitting between the quality evaluator (LLM judge) and the generator, given that both are LLMs possibly from the same family?
- Can you quantify the computational cost of one full cyclic refinement iteration (critic–adviser–reviser triplet) and estimate scalability for large corpora?
- How stable is the framework under different prompting styles or initial draft qualities? Does the cyclic loop always converge, or can it oscillate or degrade?
- Given that human evaluation was performed on 100 samples, can you report inter-annotator disagreement cases or examples where LLM and clinicians diverged significantly? I think this would make for some nice analysis in the appendix in general to showcase divergences in thinking between specialists and AI

---

> ### Author Response · Authors · 2025-11-25
> **Response to Reviewer 3q4A**
>
> 1. **Comparison with Baselines Using Real EMRs**: We sincerely appreciate your thoughtful comments. To examine whether the performance of LLM-CARe depends on access to real EMRs, we conducted an additional experiment in which LLM-CARe was provided with a real EMR as a reference when generating the initial draft.
>
> The results below show that adding a real EMR produces only minor differences in both intrinsic quality and downstream predictive performance. This indicates that the gains of LLM-CARe are driven primarily by its multi-level cyclic refinement design, and whether using a real EMR has little impact on final outcomes.
>
> **Quality Score**
> | Method          | Content Completeness | Medical Correctness | Context Consistency | Demographic Typicality | Knowledge Coverage |
> | --------------- | ------------ | ----------- | ------------------- | -------------------- | ---------------- |
> | LLM-CARe (ours)    | 91.2            | 98.6           | 93.8                   | 96.8                    | 94.1                |
> | LLM-CARe with Real EMR    | 90.3            | 97.7           | 90.5                   | 96.8                    | 97.0                |
>
> **Downstream Task Performance** (Micro Average/Macro Average)
> | Method          | Diagnosis Prediction | Examination Recommendation | Treatment Recommendation |
> | --------------- | --------- | ----------- | --------- |
> | LLM-CARe (ours)    | 82.6/82.4         | 85.3/85.2           | 76.9/74.1         |
> | LLM-CARe with Real EMR     | 83.1/82.8         | 84.9/84.2           | 77.9/75.4         |
>
> 2. A**dditional Related Work**: Thank you for the kind suggestion. We will include a citation to the recent study on LLM-based synthetic EMR generation in the Introduction of the revised manuscript.
>
> 3. **Privacy Evaluation**: We are grateful for your insightful comment regarding privacy evaluation. To assess whether our method poses any privacy risk, we evaluated privacy risk using a membership inference attack (MIA), a standard approach for testing whether generated data unintentionally reveals whether a specific record was part of its training data. In this setting, an attack accuracy of 0.50 corresponds to random guessing, so values closer to 0.50 indicate better privacy, meaning the attacker cannot distinguish training examples from non-members.
>
> As shown in the table below, LLM-CARe and LLM Direct both achieve accuracies around 0.50, which is expected because neither method has access to real EMR text during generation. LSTM and mtGAN also remain close to 0.50, likely because their model capacities are too small to memorize full clinical notes even though they were trained on real data. In contrast, MedSyn— which uses real EMRs directly as in-context examples— exhibits a noticeably higher attack accuracy, suggesting a higher risk of revealing information from specific training records.
>
> **Membership Inference Attack Accuracy**
> |Method|Attack Accuracy|
> |-|-|
> |LSTM|0.499|
> |mtGAN|0.504|
> |MedSyn|0.533|
> |LLM Direct|0.500|
> |LLM-CARe (ours)|0.504|
>
> These results indicate that LLM-CARe does not introduce detectable memorization or data leakage, consistent with its design that avoids any use of real EMR text.
>
> 4. **Concern about Evaluation Format**: Thank you for raising this important point regarding potential inductive bias arising from using downstream tasks that share structural similarity with the generation prompts. To examine whether the observed performance gains could be attributed to such textual-format alignment, we conducted an additional evaluation using data drawn from the MIMIC-IV-Note clinical database. Specifically, we selected only the subset of MIMIC discharge summaries corresponding to the same disease categories used in our study. Although these records correspond to the same disease categories, they originate from a completely different source and exhibit markedly different writing styles, structure, and documentation conventions. This setup helps mitigate the possibility that models benefit from learning superficial textual patterns.
>
> We constructed a diagnosis prediction task based on these MIMIC-derived EMRs and evaluated models trained solely on synthetic corpora. As shown below, all methods experience a significant drop in accuracy due to the distribution and stylistic mismatch—yet LLM-CARe continues to achieve the highest performance. This suggests that the advantages of LLM-CARe do not stem from alignment with the generation prompt format, but rather from improvements in the underlying clinical quality.
>
> **MIMIC-IV-Note Based Diagnosis Prediction**
> |Method|Micro Average Accuracy|Macro Average Accuracy|
> |-|-|-|
> |LSTM|0.587|0.552|
> |mtGAN|0.589|0.555|
> |MedSyn|0.584|0.571|
> |LLM Direct|0.589|0.532|
> |**LLM-CARe (ours)**|**0.606**|**0.590**|

---

> ### Author Response · Authors · 2025-11-25
> **Response to Reviewer 3q4A**
>
> 5. **Computational Cost and Scalability**: We appreciate your throughtful comments. On a single RTX 4090 GPU, producing 38k synthetic EMRs takes approximately 13 hours with direct generation, whereas LLM-CARe requires about 36 hours, which includes initial draft generation and all three refinement stages.
>
> Although the framework performs three levels of Critic–Adviser–Reviser interactions, the actual overhead is moderated by two factors. First, the outputs of these agents are much shorter than full EMRs, making each refinement step relatively lightweight. Second, if a draft already satisfies the criteria at a given stage, it bypasses subsequent agents, avoiding unnecessary iterations.
>
> Moreover, the computational cost scales linearly with the number of EMRs, and the cyclic refinement process can be parallelized across disease categories, making the framework feasible for scaling to larger corpora.
>
> 6. **Evaluation with External LLM Judges**:  Thank you for this insightful comments. To examine whether the evaluation might be biased when the generator and the judge belong to the same model family, we conducted additional assessments using two independent LLM judges. GPT-OSS-20B was selected due to its strong performance on medical tasks, and DeepSeek-V3.2-Exp (API) was included as a larger commercial model to further test robustness. For cost considerations, the DeepSeek evaluation was performed on a 10% sample of the data.
>
> As shown in the tables below, all three evaluators yield broadly consistent trends: LLM-CARe achieves the strongest performance across all intrinsic quality dimensions and maintains a clear margin over the baselines, while the ordering among the baseline methods shows only minor variation across judges. These results suggest that the conclusions of our study are not tied to any particular model family and that the observed improvements of LLM-CARe remain stable under diverse evaluation settings.
>
> **Qwen2.5-32B Evaluation**
> | Method      | Content Completeness | Medical Correctness | Context Consistency |
> | ----------- | ------------ | ----------- | ----------- |
> | LSTM        | 70.8            | 65.0           | 21.7           |
> | mtGAN       | 55.8            | 51.8           | 21.4           |
> | MedSyn      | 84.8            | 95.3           | 91.9           |
> | LLM Direct  | 77.1            | 90.7           | 87.9           |
> | **LLM-CARe**    | **91.2**            | **98.6**           | **93.8**           |
>
> **GPT-OSS-20B Evaluation**
> | Method      | Content Completeness | Medical Correctness | Context Consistency |
> | ----------- | ------------ | ----------- | ----------- |
> | LSTM        | 73.9            | 88.9           | 8.9           |
> | mtGAN       | 62.8            | 87.7           | 13.4           |
> | MedSyn      | 79.9            | 97.3           | 86.9           |
> | LLM Direct  | 81.1            | 96.7           | 70.2           |
> | **LLM-CARe**    | **93.4**            | **99.0**           | **91.3**           |
>
> **DeepSeek-V3.2-Exp Evaluation**
> | Method      | Content Completeness | Medical Correctness | Context Consistency |
> | ----------- | ------------ | ----------- | ----------- |
> | LSTM        | 69.9            | 72.5           | 2.6           |
> | mtGAN       | 57.9            | 64.0           | 0.7           |
> | MedSyn      | 80.6            | 97.5           | 85.0           |
> | LLM Direct  | 79.9            | 96.8           | 77.8           |
> | **LLM-CARe**    | **91.1**            | **98.5**           | **89.8**            |

---

> ### Author Response · Authors · 2025-11-25
> **Response to Reviewer 3q4A**
>
> 7. **Stability on Prompting Styles and Initial Draft Qualities**: We appreciate your thoughtful question regarding robustness to prompt variation and initial draft quality. To evaluate stability, we conducted two complementary experiments. First, we used GPT-5 to substantially rephrase the prompts of all agents while preserving only the high-level intent. Second, to test sensitivity to the initial draft, we replaced the LLM-Direct drafts with higher-quality drafts produced by MedSyn. The resulting quality scores and downstream task performance are shown below.
>
> **Quality Score**
> | Method          | Content Completeness | Medical Correctness | Context Consistency | Demographic Typicality | Knowledge Coverage |
> | --------------- | ------------ | ----------- | ------------------- | -------------------- | ---------------- |
> | LLM-CARe (ours)    | 91.2            | 98.6           | 93.8                   | 96.8                    | 94.1                |
> | LLM-CARe+Rephrase Prompt   | 94.1            | 98.0           | 93.8                   | 96.8                    | 95.1                |
> | LLM-CARe+Better Initial Draft    | 93.3            | 98.6           | 94.0                   | 96.8                    | 96.2                |
>
> **Downstream Task Performance** (Micro Average/Macro Average)
> | Method          | Diagnosis Prediction | Examination Recommendation | Treatment Recommendation |
> | --------------- | --------- | ----------- | --------- |
> | LLM-CARe (ours)    | 82.6/82.4         | 85.3/85.2           | 76.9/74.1         |
> | LLM-CARe+Rephrase Prompt    | 81.1/80.7         | 83.4/83.4           | 75.3/73.0         |
> | LLM-CARe+Better Initial Draft     | 82.6/81.8         | 84.7/84.4           | 76.8/73.9         |
>
> Across both settings, the numerical differences remain relatively small, and the overall pattern of results stays consistent. This stability arises from the structure of LLM-CARe’s cyclic refinement process. Because the framework decomposes quality requirements into multiple levels and lets each agent inspect or revise only one concrete criterion at a time, the system does not depend on tightly engineered prompt phrasing; as long as the criterion is stated clearly, the critic–adviser–reviser cycle can reliably identify and correct deficiencies. Besides, refinement proceeds in structured stages that strengthened quality from multiple dimensions. This layered process reduces sensitivity to the quality of the initial draft—whether the draft is relatively weak or stronger, the multi-stage refinement converges toward a similar standard of quality. Together, these properties help ensure that moderate changes in prompt wording or initial draft quality do not substantially alter the final refined EMR.

---

> ### Author Response · Authors · 2025-11-25
> **Response to Reviewer 3q4A**
>
> 8. **Convergence of the Cyclic Refinement**: Thank you for raising this important question. To ensure stable behavior of the refinement loop, our framework incorporates two complementary stopping conditions.
>
> First, refinement is performed individually for each EMR, and a draft exits the current stage as soon as it satisfies all requirements checked by the Critic. This avoids unnecessary iterations and prevents oscillation caused by repeatedly modifying already-correct content.
>
> Second, to guard against the possibility that certain requirements lie beyond the model’s ability to fully satisfy, we impose a hard upper limit of two refinement rounds per stage. This cap is based on empirical observation: performance metrics become stable after two rounds, and continued refinement provides negligible benefit while increasing computation.
>
> To confirm this empirically, we increased the iteration cap to three rounds. As shown below, both intrinsic quality and downstream task performance remain nearly unchanged relative to the two-round setting. This indicates that the refinement process has converged by the time the stopping criterion is reached, and there is no evidence of oscillation or degradation.
>
> **Quality Score**
>
> | Maximum Iterations per Stage          | Content Completeness | Medical Correctness | Context Consistency | Demographic Typicality | Knowledge Coverage |
> | --------------- | ------------ | ----------- | ------------------- | -------------------- | ---------------- |
> | 2    | 91.2            | 98.6           | 93.8                   | 96.8                    | 94.1                |
> | 3    | 91.1            | 98.1           | 93.8                   | 96.8                    | 95.6                |
>
> **Downstream Task Performance** (Micro Average/Macro Average)
>
> | Maximum Iterations per Stage          | Diagnosis Prediction | Examination Recommendation | Treatment Recommendation |
> | --------------- | --------- | ----------- | --------- |
> | 2    | 82.6/82.4         | 85.3/85.2           | 76.9/74.1         |
> | 3    | 82.9/82.9         | 85.7/85.5           | 76.7/73.9         |
>
> Overall, the combination of per-record early stopping and a global iteration cap ensures that the refinement loop converges reliably and avoids runaway revision cycles. The empirical ablation confirms that extending the loop does not materially change the final quality, indicating that the process is already converged under our stopping criteria.
>
> 9. **Examples of Disagreement**: Thank you for the comment regarding disagreement cases. Below we provide one representative case where the LLM and clinicians diverged, and one case with inter-annotator disagreement.
>
> **LLM–Clinician Disagreement**
> > Criterion: Discharge instructions should specify medication used and its dosage/usage information.\
> > Discharge Instruction: Continue oral antibiotic therapy...\
> > LLM: Correct\
> > Clinician: Wrong
>
> This case illustrates a representative pattern of divergence between clinicians and the LLM judge. When discharge instructions contain only broad statements such as “continue oral antibiotic therapy,” the LLM sometimes interprets them as sufficiently informative, whereas clinicians consider such phrasing incomplete because it lacks essential details such as the specific drug, dosage, and administration schedule. This example highlights that the model may apply a more permissive, surface-level interpretation of medication-related content, while clinical experts adhere to stricter standards grounded in actual clinical practice.
>
>
> **Inter–Clinician Disagreement**
> > The history of present illness should include the patient’s general conditions(diet, sleep, mental status, bowel/urination, weight change).\
> > History of Present Illness: The patient developed dizziness accompanied by fatigue two days ago... Appetite and sleep are normal; bowel and urination remain regular; no recent weight changeis noted.\
> > Clinician A: Correct\
> > Clinician B: Wrong
>
> In this case, the disagreement arises from how clinicians interpret whether “mental status” has been adequately described. Some clinicians consider symptoms such as dizziness and fatigue to implicitly reflect aspects of mental status—e.g., reduced alertness or decreased overall reactivity—therefore judging the general-condition description as complete. Others take a stricter view and expect an explicit statement about the patient’s orientation, consciousness, or responsiveness, and thus label the description as incomplete. This type of variation reflects differences in documentation style rather than clinical expertise, and is a common source of mild disagreement in narrative-based EMR assessment.

---

> > ### Comment · Reviewer_3q4A · 2025-11-26
> > **Please update revision**
> >
> > Thank you to the authors for their hard work throughout the rebuttal. I was high on this paper but I would appreciate if the authors would also submit a revised version of the manuscript during the author-reviewer discussion period. It appears many reviewers have asked for additional experiments and clarifications so it would be better in addition to your point by point repsonse that you updated the manuscript accordingly as well.
> >
> > I think doing so will make me quite confident that this work is ready for publication at ICLR 2026

---

> ### Author Response · Authors · 2025-12-02
> **Response Regarding Manuscript Updates**
>
> We are grateful for your suggestions and your recognition of our stage-wise multi-agent refinement framework design and extensive experiments. Based on valuable feedback from all reviewers, we have revised the manuscript to address the raised concerns, with key updates summarized below:
>
> 1. Following the reviewers’ suggestions, we further compared LLM-CARe with a generic self-refinement baseline and variants of our framework. The results indicate that the improvements in intrinsic EMR quality and downstream task performance stem from the proposed stage-wise multi-agent refinement design, rather than from generic refinement strategies or implementation details.
>
> 2. To further strengthen the rigor of the evaluation, we incorporated additional model families and an external dataset for evaluation. Across all settings, LLM-CARe consistently outperforms all baselines, further demonstrating the robustness of our approach. We also doubled the clinician-annotated sample size and observed that the clinician–LLM and inter-clinician agreement remain highly stable, validating that our original clinician evaluation was already reliable.
>
> 3. In response to reviewers’ comments, we further clarified reviewers' concerns in the manuscript—for example, providing a clearer illustration of quality principles and clarifying the construction of corpus-level reference distributions. These updates improve clarity without altering the methodology or main contributions of our work.
>
> We sincerely appreciate all reviewers for their thoughtful comments and constructive suggestions, and we hope the updated manuscript could address your concerns.

---

### Official Review · Reviewer_UAQV · 2025-11-01

**Soundness:** 3
**Presentation:** 2
**Contribution:** 2
**Rating:** 4
**Confidence:** 4

**Summary:**

This paper introduces LLM-CARe, a novel framework for generating high-quality synthetic electronic medical records using large language models. The approach employs a three-stage cyclic refinement process with specialized agents (Critic, Adviser, Reviser) that progressively improve EMR quality at corpus, section, and document levels without requiring access to real patient data during generation.

**Strengths:**

1.  LLM-CARe aims to generate realistic EMRs without relying on real patient records, addressing a critical privacy barrier in healthcare AI. This approach could enable large-scale data creation for medical model training without exposing sensitive information.
2. The paper conducts detailed ablation studies isolating the Critic, Adviser, and Reviser components, as well as corpus-, section-, and document-level refinements. These experiments clearly demonstrate that each module contributes meaningfully to the overall EMR quality improvements, validating the necessity of the multi-agent architecture.
3. Evaluation is multi-layered—combining automatic LLM-based judging, human clinician assessment, and downstream task performance
4. The framework is tested across several LLM backbones (Qwen 2, LLaMA 3.1, Meditron, R1) and maintains performance advantages on all of them, indicating robustness to model architecture and training data differences.
5. The paper includes detailed prompts, backbone configurations, and ablation setups, signaling commitment to transparent and reproducible research

**Weaknesses:**

1. The paper’s structure is fragmented: key definitions, evaluation criteria, and implementation details are scattered across main text and appendices. This hampers readability and makes it difficult to follow the end-to-end methodology.
2. The five quality dimensions (content completeness, medical correctness, context consistency, demographic typicality, knowledge coverage) are not well defined in the main text. Critical terms — e.g., “major” vs “minor” symptoms — are undefined.Additionally, the paper provides insufficient detail about baseline methods (particularly MedSyn, described in only one sentence )
3. The introduction claims to address “insufficient coverage of less typical clinical cases,” yet the dataset construction retains only common diagnoses (≥500 records) and explicitly excludes rare conditions. This directly contradicts the motivation presented by the authors and undermines the paper’s central claim.
4. The clinician validation is small (100 records, 4 clinicians) and the paper reports aggregate clinician–LLM agreement without per-criterion reliability (e.g., per-criterion Cohen’s κ, confusion matrices). Given the paper relies on an LLM judge, stronger human validation and per-criterion calibration are needed.
5. The corpus-level alignment step matches synthetic data to a reference distribution. The paper never specifies whether 𝑇\_𝑑  is computed from public statistics, synthetic approximations, or the authors’ real EMR dataset. If it is derived from real EMRs, it contradicts their claim that the generation “requires no real EMRs”. The source and exact construction of this distribution must be stated explicitly.
6. Aligning synthetic data to a reference distribution derived from real data risks reproducing existing demographic or clinical biases. The paper lacks subgroup/fairness analyses to evaluate whether corpus-level alignment amplifies or mitigates such biases.
7. The experiments predominantly use a single model family (Qwen) for generation, judging, and downstream evaluation, which risks correlated errors and vendor-specific bias. This open the possibility that performance gains may be partially due to vendor/model-specific bias rather than general methodological benefit.
8. Missing critical details such as exact sample sizes for each downstream task.
9. The authors provide no statistics on EMR length, number of sections, or content depth, nor do they compare these properties between synthetic and real EMRs. Without such analysis, claims of realism and practical utility are unsupported.
10. Medical correctness is assessed via surface-level alignment checks rather than deeper clinical plausibility. The evaluation does not test whether records reflect realistic multi-disease interactions or plausible event chronology.

**Questions:**

1. Please clarify whether the corpus-level reference distributions used for alignment are derived from real EMRs, public statistics, or synthetic approximations. If real EMRs were used, specify exactly which aggregate statistics (e.g., age, gender, disease frequency) were extracted.
2. Report per-criterion Cohen’s Kappa values between clinician and LLM judgments, as well as inter-clinician Kappas.
Include confusion matrices for each quality dimension on the clinician-labeled samples.
3. Why was exact string matching chosen instead of concept-level or semantic matching (e.g., UMLS normalization or fuzzy embedding similarity)?
4. Provide comparative results using a semantic approach or at least an analysis quantifying the under-count due to exact matching.
5. Supply (a) the distributions of age, gender, and disease before and after each filtering stage, and (b) an analysis of how the ≥ 500-record disease cutoff affects downstream generalization to less-common conditions.
6. Provide paired statistical significance tests comparing LLM-CARe with each baseline on primary metrics. Include per-disease performance tables or aggregates grouped by disease frequency.
7. Consider evaluating with an independent LLM judge (from a different vendor), expanding clinician evaluation beyond 100 EMRs, and including edge or rare cases.
8. Provide subgroup analyses (e.g., by gender, age bracket, or ethnicity if available) for both EMR quality metrics and downstream task outcomes.

---

> ### Author Response · Authors · 2025-11-25
> **Response to Reviewer UAQV**
>
> 1. **Comments on Writing and Sturcture**: We sincerely appreciate your detailed suggestions. In the original submission, we focused the main text on presenting the core framework and empirical findings, which led us to place several definitions, clinical explanations, and implementation details in the appendix due to space constraints. Following your suggestions, we will improve the manuscript by adding illustrative examples for the quality dimensions, expanding the descriptions of baseline methods, and providing clearer references between the main text and the appendix to enhance the readability and coherence of the overall methodology.
>
> 2. **Clarification on “Less Typical Clinical Cases”**: We are grateful for your thoughtful comments. Our original wording may not have clearly conveyed what we meant by “less typical clinical cases.” In this work, the term refers to less common clinical presentations within a given disease, rather than rare diseases. Since all methods in our comparison (including ours and the baselines) are designed to generate EMRs conditioned on a specified diagnosis, the coverage of disease categories can be controlled directly. What we observed, however, is that baseline models tend to produce only the most typical or frequently seen manifestations of the given disease—for example, generating the same high-frequency symptom combinations or standard treatment paths—while failing to cover valid but less common presentations or management choices.
>
> This is the coverage gap we aim to address. Through corpus-level refinement, LLM-CARe explicitly adjusts the synthetic dataset toward broader coverage of clinical knowledge and demographic patterns, which encourages inclusion of these less typical but clinically valid scenarios within each disease category. Our experiments show that the refined corpus indeed covers a more diverse set of symptoms, examinations, and treatment patterns than the baselines.
>
> Regarding disease filtering, we retain only diagnoses with at least 500 real EMRs to ensure stable and reliable downstream evaluation. Rare diseases with very limited real samples lead to highly volatile test-set metrics, making comparative evaluation unreliable. Although our current experiments focus on intra-disease diversity under well-supported diagnoses, we note that the framework could in principle be extended to rare diseases as well, which we consider a promising direction for future work.
>
> To avoid ambiguity, we will revise the manuscript to clearly state that “insufficient coverage of less typical clinical cases” focuses on intra-disease diversity rather than rare diseases.

---

> ### Author Response · Authors · 2025-11-25
> **Response to Reviewer UAQV**
>
> 3. **Realiability of Clinician Evaluation**: We would like to express our gratitude to your constructive feedback. To strengthen human validation and assess the stability of clinician–LLM agreement, we expanded the clinician evaluation beyond the initial 100 records. In sampling the additional data, we stratified diseases by frequency into multiple groups and sampled an equal number of cases from each group, ensuring that both high-frequency and lower-frequency situations were well represented. We likewise balanced the samples across generation methods, gender, and age groups to ensure comprehensive coverage.
>
> Clinicians then annotated two additional sets of 50 EMRs each, producing subsets of 100, 150, and 200 annotated records. We recomputed clinician–LLM agreement at each subset size. The agreement values remained consistently close across the three settings, while the confidence intervals naturally narrowed as the sample size increased. Across all sample sizes, the clinician–LLM agreement (Cohen’s κ) remained above 0.8, indicating strong alignment between human and model judgments. The inter-clinician agreement (Fleiss’s κ) was above 0.9 in all settings, reflecting very high consistency among clinicians themselves. These results suggest that both the human annotations and the model-based evaluations are stable and reliable.
>
> **Extanded Clinician Evaluation**
> | # Samples | Clinician-LLM Agreement (Cohen’s κ) | 95% CI (Cohen’s κ) |  Inter-Clinician Agreement (Fleiss’s κ) | 95% CI (Fleiss’s κ) |
> |----------|-----------|-------------|----------|--------------|
> | 100      | 0.837         | [0.804, 0.868]           | 0.947        | [0.928, 0.964]            |
> | 150      | 0.842         | [0.816， 0.868]           | 0.948        | [0.933, 0.962]            |
> | 200      | 0.833         | [0.809, 0.855]           | 0.950        | [0.937, 0.962]            |
>
> For the per-criterion analysis, both clinician–LLM agreement and inter-clinician agreement were generally high across the 200 annotated samples. Most criteria showed κ values above 0.7–0.8 for clinician–LLM agreement and above 0.9 for inter-clinician agreement, indicating stable and reliable assessments. The two medication-related criteria exhibited noticeably lower clinician–LLM agreement, which likely reflects the current limitations of LLMs in handling detailed medication knowledge. The criterion “Diagnosis matches the patient’s gender” also showed a lower κ, largely due to extreme label imbalance (188 true positive with only a few disagreements), a setting in which Cohen’s κ is known to produce low values.
>
> **Per-Criterion Aggrement**
> | Criterion | Clinician-LLM Agreement (Cohen’s κ) | 95% CI (Cohen’s κ) |  Inter-Clinician Agreement (Fleiss’s κ) | 95% CI (Fleiss’s κ) |
> |----------|-----------|-------------|----------|--------------|
> |Chief complaint states reason for admission|0.818|[0.496, 1.000]|1.000|[1.000, 1.000]|
> |Chief complaint includes onset time|0.843|[0.645, 1.000]|1.000|[1.000, 1.000]|
> |History of present illness lists major symptoms and onset time|0.882|[0.754, 0.975]|1.000|[1.000, 1.000]|
> |History of present illness describes acuity of onset|0.728|[0.616, 0.828]|0.910|[0.844, 0.966]|
> |History of present illness mentions possible causes|0.783|[0.684, 0.874]|0.989|[0.963, 1.000]|
> |History of present illness includes all general conditions|0.950|[0.900, 0.990]|0.959|[0.917, 0.990]|
> |Hospital course includes auxiliary examinations or laboratory examinations|0.862|[0.775, 0.935]|0.975|[0.934, 1.000]|
> |Hospital course includes treatment interventions|0.987|[0.957, 1.000]|1.000|[1.000, 1.000]|
> |Discharge instruction includes medication dosage and usage|0.548|[0.395, 0.687]|0.916|[0.845, 0.974]|
> |Diagnosis matches the patient’s gender|0.638|[0.315, 0.884]|1.000|[1.000, 1.000]|
> |Symptoms in chief complaint align with diagnosis|0.800|[0.651, 0.922]|0.911|[0.811, 0.981]|
> |Symptoms in history of present illness align with diagnosis|0.733|[0.626, 0.831]|0.889|[0.816, 0.948]|
> |Examinations in hospital course align with diagnosis|0.710|[0.606, 0.807]|0.877|[0.804, 0.939]|
> |Medications in discharge instructions align with diagnosis|0.434|[0.280, 0.583]|0.780|[0.680, 0.869]|
> |Symptoms in chief complaint are consistent with those in history of present illness|0.884|[0.812, 0.946]|0.990|[0.968, 1.000]|
> |Onset time in chief complaint is consistent with that in history of present illness|0.869|[0.799, 0.930]|0.980|[0.950, 1.000]|
> |Affected site in history of present illness is consistent with the site of examination or treatment in hospital course|0.858|[0.780, 0.928]|0.990|[0.969, 1.000]|

---

> ### Author Response · Authors · 2025-11-25
> **Response to Reviewer UAQV**
>
> **Confusion Matric on Content Completeness**
> ||LLM: True|LLM: False|
> |-|-|-|
> |Human: True|1265|40|
> |Human: False|55|440|
>
> **Confusion Matric on Medical Correctness**
> ||LLM: True|LLM: False|
> |-|-|-|
> |Human: True|746|30|
> |Human: False|68|156|
>
> **Confusion Matric on Context Consistency**
> ||LLM: True|LLM: False|
> |-|-|-|
> |Human: True|309|20|
> |Human: False|18|253|
>
> The confusion matrices show that, for all three quality dimensions, most samples fall into the True Positive and True Negative cells, indicating broad alignment between clinician and LLM judgments. Across dimensions, False Positives are more frequent than False Negatives, suggesting that the LLM is, on average, somewhat less strict than clinicians. The disagreement cases are relatively limited and not concentrated in any particular dimension, supporting the conclusion that the LLM’s evaluations are generally consistent with human assessments without exhibiting systematic bias.
>
> 4. **Source of Corpus-Level Distribution**: We appreciate your thoughtful comments. Our original wording may not have clearly conveyed what we meant by “requiring no real EMRs.” What we intended to convey is that unlike baselines such as LSTM, mtGAN, or MedSyn—which rely on real EMR text for training or prompting—our method does not use any real clinical narratives during generation. The corpus-level reference distribution used in LLM-CARe is derived only from aggregate statistics computed over the training set, including the age distribution, gender proportion, and the frequency of medical concepts we concerned (e.g. symptoms, medications). These are population-level summaries and do not contain any information tied to specific patient records (e.g., no per-record age, symptom combinations), so they do not expose individual privacy.
>
> We note that in real-world settings, raw EMR text typically cannot leave hospital infrastructure due to privacy regulations, making it difficult to use directly. In contrast, aggregate statistics do not expose individual privacy and can be computed within the hospital and safely used outside the protected environment. Moreover, such demographic and prevalence distributions can often be obtained from publicly available epidemiological reports or health statistics, without requiring access to EMR text at all. For this reason, our approach—relying only on statistical distributions rather than EMR text—is easier to adopt in practical deployments.
>
> We will clarify this point in the revised manuscript and explicitly described the source and construction of the reference distributions.

---

> ### Author Response · Authors · 2025-11-25
> **Response to Reviewer UAQV**
>
> 5. **Subgroup Analysis**: We appreciate the reviewer’s insightful comments on potential demographic or clinical biases. Indeed, real-world EMR distributions may contain imbalance, and methods that train on or prompt from real EMRs may inherit these biases without explicit control. In contrast, LLM-CARe offers more flexibility: our corpus-level stage uses explicit distributional targets, meaning that if the underlying statistics are biased, the target distribution can be manually corrected before generation (for example, adjusting a skewed male–female ratio to 1:1). Such manual debiasing is difficult to achieve in baseline methods that depend directly on real EMR text.
>
> To assess whether our approach amplifies or mitigates subgroup disparities, we performed a subgroup analysis along gender and age (using standard WHO brackets 0–17, 18–44, 45–64, 65+). Because Demographic Typicality itself evaluates demographic distribution alignment, we excluded it from the stratified analysis. The results show that LLM-CARe consistently achieves the highest intrinsic quality scores and downstream task performance across most of subgroups, and that the variation across subgroups is comparable to that of the baselines.
>
> We additionally observed that all methods exhibit lower downstream task performance in the 65+ age group—likely reflecting the complexity and heterogeneity of cases in this demographic. Nonetheless, LLM-CARe still achieves the best performance among all methods within this challenging subgroup, indicating that our corpus-level alignment and multi-level refinement help maintain robustness even in difficult populations.
>
> These findings suggest that our framework does not reinforce demographic biases; instead, by allowing explicit distribution control, it can better support balanced dataset construction compared with baselines that rely directly on raw real-world distributions.
>
> **Quality Score by Gender**
> | Gender | Method | Content Completeness | Medical Correctness | Context Consistency | Knowledge Coverage |
> |-| --------------- | ------------ | ----------- | ------------------- | -------------------- |
> |Female| LSTM | 71.9 | 65.0 | 21.9 | 70.3 |
> |Female| mtGAN | 55.4 | 51.7 | 21.2 | 76.2 |
> |Female| MedSyn | 85.2 | 95.6 | 91.7 | 84.4 |
> |Female| LLM Direct | 76.9 | 90.8 | 88.0 | 73.7 |
> |Female| **LLM-CARe (ours)** | **91.1** | **98.7** | **93.9** | **93.8** |
> |Male| LSTM | 69.8 | 65.0 | 21.5 | 70.3 |
> |Male| mtGAN | 56.2 | 51.9 | 21.6 | 76.2 |
> |Male| MedSyn | 84.4 | 95.0 | 92.2 | 84.4 |
> |Male| LLM Direct | 77.2 | 90.7 | 87.9 | 73.8 |
> |Male| **LLM-CARe (ours)** | **91.3** | **98.5** | **93.7** | **94.0** |
>
> **Downstream Task Performance by Gender** (Micro Average/Macro Average)
>
> | Gender | Method | Diagnosis Prediction | Examination Recommendation | Treatment Recommendation |
> |-| --------------- | --------- | ----------- | --------- |
> |Female| LSTM | 74.1/72.2 | 75.5/76.2 | 54.7/49.3 |
> |Female| mtGAN | 82.1/80.5 | 72.5/73.6 | 57.1/52.3 |
> |Female| MedSyn | 82.2/81.5 | 82.7/82.0 | 73.9/71.4 |
> |Female| LLM Direct | 82.1/81.3 | 64.6/65.3 | 59.8/58.4 |
> |Female| *LLM-CARe (ours)* | **83.0**/**82.0** | **85.3**/**85.3** | **76.1**/**73.7** |
> |Male| LSTM | 73.9/72.3 | 75.8/77.1 | 58.5/53.4 |
> |Male| mtGAN | 81.7/79.8 | 72.3/73.9 | 59.9/55.0 |
> |Male| MedSyn | 81.2/80.4 | 83.0/82.5 | 75.1/71.7 |
> |Male| LLM Direct | 81.5/81.2 | 64.2/65.3 | 61.9/59.9 |
> |Male| **LLM-CARe (ours)** | **82.2**/**81.4** | **85.3**/**85.4** | **77.7**/**75.3** |

---

> ### Author Response · Authors · 2025-11-25
> **Response to Reviewer UAQV**
>
> **Quality Score by Age Group**
>
> | Age Group | Method | Content Completeness | Medical Correctness | Context Consistency | Knowledge Coverage |
> |-| --------------- | ------------ | ----------- | ------------------- | -------------------- |
> |0-17| LSTM | 72.7 | 62.2 | 25.1 | 69.8 |
> |0-17| mtGAN | 55.4 | 49.5 | 23.8 | 75.7 |
> |0-17| MedSyn | 85.9 | 97.0 | 94.1 | 83.4 |
> |0-17| LLM Direct | 76.3 | 91.1 | 89.4 | 73.3 |
> |0-17| **LLM-CARe (ours)** | **91.7** | **97.8** | **97.2** | **93.7** |
> |18-44| LSTM | 69.1 | 63.6 | 20.4 | 70.1 |
> |18-44| mtGAN | 56.1 | 50.9 | 21.0 | 76.0 |
> |18-44| MedSyn | 84.7 | 95.4 | 92.5 | 83.9 |
> |18-44| LLM Direct | 77.0 | 90.5 | 88.6 | 73.5 |
> |18-44| **LLM-CARe (ours)** | **91.1** | **98.6** | **94.7** | **93.8** |
> |45-64| LSTM | 70.2 | 65.1 | 21.3 | 70.3 |
> |45-64| mtGAN | 55.9 | 52.3 | 21.1 | 76.2 |
> |45-64| MedSyn | 85.1 | 95.3 | 91.7 | 84.4 |
> |45-64| LLM Direct | 77.2 | 90.7 | 87.8 | 73.8 |
> |45-64| **LLM-CARe (ours)** | **91.2** | **98.7** | **93.4** | **94.0** |
> |65+| LSTM | 71.8 | 66.3 | 22.0 | 70.2 |
> |65+| mtGAN | 55.7 | 52.4 | 21.3 | 76.1 |
> |65+| MedSyn | 83.6 | 94.8 | 91.6 | 84.0 |
> |65+| LLM Direct | 76.8 | 90.8 | 87.4 | 73.5 |
> |65+| **LLM-CARe (ours)** | **91.1** | **98.7** | **93.0** | **93.8** |
>
> **Downstream Task Performance by Age Group** (Micro Average/Macro Average)
>
> | Age Group | Method | Diagnosis Prediction | Examination Recommendation | Treatment Recommendation |
> |-| --------------- | --------- | ----------- | --------- |
> |0-17| LSTM | 78.9/77.7 | 78.0/78.6 | 56.6/44.6 |
> |0-17| mtGAN | **86.4**/81.9 | 74.5/75.6 | 62.3/50.2 |
> |0-17| MedSyn | 84.5/**84.3** | 82.6/83.7 | 71.8/**73.4** |
> |0-17| LLM Direct | 83.4/82.1 | 64.7/70.1 | 68.7/60.8 |
> |0-17| **LLM-CARe (ours)** | 85.5/83.3 | **87.9**/**88.4** | **79.5**/72.6 |
> |18-44| LSTM | 79.1/72.5 | 76.7/77.6 | 43.6/48.0 |
> |18-44| mtGAN | 85.6/80.4 | 73.2/75.6 | 48.0/52.3 |
> |18-44| MedSyn | 85.9/**83.1** | 83.8/83.1 | 72.7/74.8 |
> |18-44| LLM Direct | **86.3**/82.7 | 65.9/68.6 | 56.9/62.4 |
> |18-44| **LLM-CARe (ours)** | **86.3**/82.9 | **85.7**/**85.9** | **73.8**/**76.0** |
> |45-64| LSTM | 75.3/72.9 | 75.7/76.3 | 56.1/49.5 |
> |45-64| mtGAN | 82.5/78.7 | 72.6/73.6 | 58.0/52.3 |
> |45-64| MedSyn | 83.0/82.0 | 83.0/82.4 | 75.5/71.8 |
> |45-64| LLM Direct | 83.0/**82.3** | 64.7/66.2 | 60.8/59.3 |
> |45-64| **LLM-CARe (ours)** | **83.8**/81.9 | **85.0**/**85.1** | **77.6**/**74.2** |
> |65+| LSTM | 69.2/70.1 | 74.9/76.1 | 60.6/52.5 |
> |65+| mtGAN | 78.5/78.6 | 71.5/73.1 | 61.3/54.2 |
> |65+| MedSyn | 77.8/79.2 | 82.5/81.8 | 74.6/69.9 |
> |65+| LLM Direct | 78.1/79.7 | 63.6/64.6 | 60.9/57.8 |
> |65+| **LLM-CARe (ours)** | **79.0**/**79.9** | **84.9**/**84.9** | **76.9**/**73.2** |

---

> ### Author Response · Authors · 2025-11-25
> **Response to Reviewer UAQV**
>
> 6. **LLM-as-Judge Choice**: We appreciate your concern about the possibility of vendor-specific bias, given that our main experiments used the Qwen family for generation, evaluation, and downstream modeling. To address this, we conducted additional evaluations with two independent LLM judges from different model families. GPT-OSS-20B was selected for its strong performance on medical reasoning benchmarks, and DeepSeek-V3.2-Exp (API) was added as a larger commercial model to further test robustness. For cost considerations, the DeepSeek evaluation was performed on 10% sampled data.
>
> The results are summarized in the tables below. While the absolute scores vary across evaluators, the overall conclusions remain stable: LLM-CARe achieves the highest scores under all three judges, and the relative ranking among the baseline methods differs only slightly across models. These findings indicate that our quality improvements are not tied to a specific model family and that the superiority of LLM-CARe is consistent across heterogeneous evaluators.
>
> **Quality Score with Qwen2.5-32B**
> | Method      | Content Completeness | Medical Correctness | Context Consistency |
> | ----------- | ------------ | ----------- | ----------- |
> | LSTM        | 70.8            | 65.0           | 21.7           |
> | mtGAN       | 55.8            | 51.8           | 21.4           |
> | MedSyn      | 84.8            | 95.3           | 91.9           |
> | LLM Direct  | 77.1            | 90.7           | 87.9           |
> | **LLM-CARe**    | **91.2**            | **98.6**           | **93.8**           |
>
> **Quality Score with GPT-OSS-20B**
> | Method      | Content Completeness | Medical Correctness | Context Consistency |
> | ----------- | ------------ | ----------- | ----------- |
> | LSTM        | 73.9            | 88.9           | 8.9           |
> | mtGAN       | 62.8            | 87.7           | 13.4           |
> | MedSyn      | 79.9            | 97.3           | 86.9           |
> | LLM Direct  | 81.1            | 96.7           | 70.2           |
> | **LLM-CARe**    | **93.4**            | **99.0**           | **91.3**           |
>
> **Quality Score with DeepSeek-V3.2-Exp**
> | Method      | Content Completeness | Medical Correctness | Context Consistency |
> | ----------- | ------------ | ----------- | ----------- |
> | LSTM        | 69.9            | 72.5           | 2.6           |
> | mtGAN       | 57.9            | 64.0           | 0.7           |
> | MedSyn      | 80.6            | 97.5           | 85.0           |
> | LLM Direct  | 79.9            | 96.8           | 77.8           |
> | **LLM-CARe**    | **91.1**            | **98.5**           | **89.8**            |
>
> In addition, we replaced the backbone for downstream task evaluation and trained all models using Llama-3.2-1B-Instruct to further assess robustness under a different model architecture. The relative ordering of the methods remains stable: LLM-CARe again achieves the best performance across diagnosis prediction, examination recommendation, and treatment recommendation. This cross-model consistency indicates that the downstream gains of LLM-CARe do not depend on characteristics of the Qwen model family and instead stem from the proposed multi-level refinement framework itself.
>
> **Downstream Task Performance with Llama-3.2-1B-Instruct** (Micro Average/Macro Average)
> | Method          | Diagnosis Prediction | Examination Recommendation | Treatment Recommendation |
> | --------------- | --------- | ----------- | --------- |
> | LSTM  | 68.5/67.1         | 74.7/75.4           | 55.6/48.6         |
> | mtGAN  | 78.8/77.7         | 72.8/73.6           | 58.9/53.1         |
> | MedSyn  | 77.2/77.8         | 82.4/81.7           | 72.0/68.7         |
> | LLM Direct  | 78.9/79.0         | 61.9/62.8           | 59.4/57.0         |
> | **LLM-CARe (ours)**    | **80.1**/**79.8**         | **83.3**/**83.2**           | **74.7**/**71.8**         |

---

> ### Author Response · Authors · 2025-11-25
> **Response to Reviewer UAQV**
>
> 7. **Sample Size for Downstream Tasks**: We appreciate your detailed comments to the setting of our experiment. We count the exact sample sizes for all downstream tasks, summarized in the table below. In the diagnosis prediction task, each EMR corresponds to exactly one question, so all methods share the same number of samples (30,308). In contrast, the examination and treatment recommendation tasks generate one question per extracted examination or treatment entity. Because an EMR could contain multiple such entities, the total number of questions varies across methods.
>
> As shown in the table, LSTM and mtGAN produce fewer examination and treatment items due to their limited generation quality and smaller model capacity. Among LLM-based methods, LLM-CARe and MedSyn have comparable sample sizes, whereas LLM Direct produces noticeably more items. Despite this larger number of questions, LLM Direct still performs substantially worse, largely because its generated records exhibit low diversity and overproduce high-frequency concepts while failing to cover clinically diverse cases. This indicates that LLM-CARe’s improvements are not the result of simply generating more entities, but rather stem from producing higher-quality and more diverse EMRs under similar data scales.
>
> **Sample Size for Downstream Tasks**
> | Method          | Diagnosis Prediction | Examination Recommendation | Treatment Recommendation |
> | --------------- | --------- | ----------- | --------- |
> | LSTM  | 38308         | 154599           | 64662         |
> | mtGAN  | 38308         | 187676           | 70296         |
> | MedSyn  | 38308         | 212710           | 97305         |
> | LLM Direct  | 38308         | 266927           | 139107         |
> | LLM-CARe (ours)    | 38308        | 206951           | 102658        |
> |Test Set| 38308 | 346045 | 110494 |
>
> 8. **Statistics of EMR**: Thank you for your thoughtful comments. All methods in our study generate the same four sections, and we report the token length of each section in the table below. Real EMRs are substantially longer due to the rich descriptive details physicians include during documentation. Among baselines, LSTM, mtGAN, and MedSyn produce relatively longer notes because they are training or prompting with real EMR text, though they are still shorter of real documents. In contrast, LLM-CARe does not use any real EMR text, yet its multi-stage refinement process significantly enriches each section and produces much longer and more informative records than the initial LLM Direct drafts. This demonstrates that iterative quality control effectively supplements content even without exposure to real narratives, and enables LLM-CARe to better approximate the structural characteristics of real EMRs. Further exploring how to approach real-world documentation patterns without using protected clinical text is a valuable direction for future work.
>
> **Number of Tokens Per Section**
> | Method          | Chief Complaint | History of Present Illness | Hospital Course | Discharge Insturction | Total |
> | --------------- | --------- | ----------- | --------- |-|-|
> |LSTM|13|227|131|88|458|
> |mtGAN|24|202|166|101|493|
> |MedSyn|14|185|152|138|488|
> |LLM Direct|14|90|82|76|262|
> |LLM-CARe (ours)|15|125|119|116|375|
> |Real EMR|12|231|291|168|702|
>
> 9. **Scope of Medical Correctness Evaluation**: We appreciate your thoughtful suggestions. The issues raised—such as multi-disease interactions and event chronology—are indeed important in real clinical settings. In this work, however, we follow the setup commonly adopted in prior EMR-generation research, where each record is conditioned on a single diagnosis, and the goal is to ensure basic medical correctness within that scope. Even under this simpler setting, existing baselines show notable errors, so we focus on addressing these fundamental issues before extending to more complex multi-disease interactions. Besides, the EMRs used in our study are narrative clinical notes that summarize key information from admission to discharge. Although they contain coarse temporal information, they do not provide fine-grained event sequences. We observe that current methods already struggle with basic cross-section consistency within this narrative structure. As such, deeper assessments of multi-disease interplay or event-level timelines are beyond the scope of the current data and setup. We agree that these are valuable directions and we view exploring richer clinical plausibility as an important area for future work.

---

> ### Author Response · Authors · 2025-11-25
> **Response to Reviewer UAQV**
>
> 10. **Exact vs. Semantic Matching**: We are grateful for your insightful comments. We agree semantic matching could capture additional valid expressions beyond exact lexical forms, so we conducted an additional analysis using a semantic strategy. In this setting, we encode both the target clinical concepts and EMR text spans using a BERT-base encoder and treat a pair as matched when their cosine similarity exceeds a selected threshold.
>
> The results are shown in the table below. Semantic matching increases the absolute scores for all methods, as expected, but the changes are modest and—most importantly—the relative ranking of all models remains unchanged, with LLM-CARe consistently achieving the highest coverage. We believe this stability arises because many concepts in our knowledge set naturally include multiple synonymous expressions, allowing exact matching to capture a large portion of valid mentions.
>
> Given that semantic matching introduces substantially higher computational cost while yielding similar comparative conclusions, we use exact matching as the main metric in the paper and report the semantic-matching analysis in the revision for completeness.
>
> **Comparison of Matching Strategy on Knowledge Coverage**
> |Method|Exact Match|Semantic Match|
> |-|-|-|
> |LSTM|70.4|75.1|
> |mtGAN|76.3|83.3|
> |MedSyn|84.5|87.4|
> |LLM Direct|73.9|78.8|
> |**LLM-CARe (ours)**|**94.1**|**95.0**|
>
> 11. **Distribution Across Filtering Stages**: We appreciate your comments on details regarding dataset construction. We report the distributions of age, gender, and disease categories before and after each filtering stage, summarized in the tables below. For gender and age, the distributions remain largely stable across all stages, indicating that the preprocessing steps do not introduce noticeable demographic bias. In the original dataset, there are 21,902 ICD-coded diseases. Length filtering reduces this to 15,863 diseases. We then retain only diseases with at least 500 records, resulting in 302 diseases. This choice is motivated by the need to evaluate downstream models on held-out real EMRs for each disease category. If a disease contains only a small number of samples, the test-set performance becomes extremely unstable and statistically unreliable. To ensure robust and meaningful evaluation, we therefore exclude low-frequency diseases from the experiments.
>
> **Gender Distribution across Filtering Stages**
> |Filtering Stage|Female|Male|
> |-|-|-|
> |Original|52.1%|47.9%|
> |After Length Filtering|51.2%|48.8%|
> |After Disease Filtering|50.1%|49.9%|
>
> **Age Distribution across Filtering Stages**
> |Filtering Stage|0-17|18-44|45-64|65+|
> |-|-|-|-|-|
> |Original|8.7%|21.6%|36.5%|33.2%|
> |After Length Filtering|8.9%|19.6%|37.2%|34.3%|
> |After Disease Filtering|8.8%|17.2%|37.1%|36.8%|
>
> 12. **Statistical Significance**: To examine whether the performance improvements of LLM-CARe are statistically significant, we performed two-proportion z-tests on the intrinsic quality metrics. Because each method independently generates its own set of synthetic EMRs, the samples are not paired, and independent-proportion testing is appropriate. As shown in the table below, LLM-CARe achieves significantly higher content completeness, medical correctness, and context consistency than all baselines (p < 0.001 throughout).
>
> **Two-Proportion Z-Test for Quality Metrics (p-values, LLM-CARe vs. baseline)**
> | Baseline      | Content Completeness | Medical Correctness | Context Consistency |
> | ----------- | ------------ | ----------- | ----------- |
> | LSTM        | <0.001            | <0.001           | <0.001           |
> | mtGAN       | <0.001            | <0.001           | <0.001           |
> | MedSyn      | <0.001            | <0.001           | <0.001           |
> | LLM Direct  | <0.001            | <0.001           | <0.001           |
>
> For downstream tasks, all methods make predictions on the same set of real EMRs, making the evaluation paired. Therefore, we applied McNemar’s test, which is appropriate for paired binary outcomes. As shown in the table below, LLM-CARe significantly outperforms all baselines on downstream tasks; nearly all p-values are < 0.001, confirming that the improvements are not due to random variation.
>
> **Paired McNemar Test for Downstream Tasks (p-values, LLM-CARe vs. baseline)**
> | Baseline          | Diagnosis Prediction | Examination Recommendation | Treatment Recommendation |
> | --------------- | --------- | ----------- | --------- |
> | LSTM  | <0.001         | <0.001           | <0.001         |
> | mtGAN  | <0.001         | <0.001           | <0.001         |
> | MedSyn  | <0.001         | <0.001           | <0.001         |
> | LLM Direct  | 0.005         | <0.001           | <0.001         |
>
> These results show that LLM-CARe’s advantages over the baselines are statistically robust for both intrinsic quality and downstream clinical utility.

---

> ### Author Response · Authors · 2025-11-25
> **Response to Reviewer UAQV**
>
> 13. **Performance Across Disease-Frequency Groups**: We sincerely appreciate your suggestion to analyze performance across diseases with different frequencies. To address this, we grouped the 302 retained diagnoses into four frequency strata based on their prevalence in the real dataset (top 25%, 25–50%, 50–75%, and bottom 25%), and evaluated intrinsic quality and downstream performance within each group.
>
> As expected, all methods show somewhat better results on more common diseases, but the differences across groups are not substantial. LLM-CARe achieves strong performance in every frequency group, demonstrating that the method is effective not only for high-frequency diseases but also for less frequent ones within the dataset.
>
> **Quality Score by Disease Frequency Group**
>
> | Disease Frequency Group | Method | Content Completeness | Medical Correctness | Context Consistency | Demographic Typicality | Knowledge Coverage |
> |-|-| --------------- | ------------ | ----------- | ------------------- | -------------------- |
> |Top 25%| LSTM |66.8|71.4|22.6|94.0|72.5|
> |Top 25%| mtGAN |54.2|56.4|22.7|94.3|78.6|
> |Top 25%| MedSyn |85.3|95.9|92.4|84.9|83.8|
> |Top 25%| LLM Direct |90.7|77.2|87.6|73.3|77.1|
> |Top 25%| **LLM-CARe (ours)** |91.3|98.6|93.9|97.3|94.9|
> |25%-50%| LSTM |63.7|70.2|21.3|93.0|72.6|
> |25%-50%| mtGAN |50.3|56.0|20.9|93.6|78.3|
> |25%-50%| MedSyn |84.5|94.6|91.5|84.8|83.2|
> |25%-50%| LLM Direct |90.6|77.0|87.8|75.3|78.4|
> |25%-50%| **LLM-CARe (ours)** |91.1|98.4|93.3|96.9|94.3|
> |50%-75%| LSTM |64.3|70.6|21.0|92.9|67.5|
> |50%-75%| mtGAN |50.8|55.0|20.6|93.1|73.7|
> |50%-75%| MedSyn |84.4|94.7|91.1|83.2|85.6|
> |50%-75%| LLM Direct |90.8|77.0|87.9|73.3|78.1|
> |50%-75%| **LLM-CARe (ours)** |91.3|98.7|93.3|96.8|93.4|
> |Bottom 25%| LSTM |63.9|70.6|21.2|93.3|68.8|
> |Bottom 25%| mtGAN |50.1|55.3|20.4|93.3|74.5|
> |Bottom 25%| MedSyn |84.7|95.7|92.3|83.5|85.4|
> |Bottom 25%| LLM Direct |90.6|77.0|88.5|73.8|77.2|
> |Bottom 25%| **LLM-CARe (ours)** |91.1|98.7|94.7|96.3|93.7|
>
> **Downstream Task Performance by Disease Frequency Group** (Micro Average/Macro Average)
>
> | Disease Frequency Group | Method | Diagnosis Prediction | Examination Recommendation | Treatment Recommendation |
> |-| --------------- | --------- | ----------- | --------- |
> |Top 25%| LSTM | 78.0/77.5 | 76.9/77.6 | 62.1/54.0 |
> |Top 25%| mtGAN |85.4/84.7|73.9/74.7|63.5/57.0|
> |Top 25%| MedSyn |82.6/83.2|83.9/83.2|77.9/74.3|
> |Top 25%| LLM Direct |82.2/82.8|65.0/66.4|63.9/62.0|
> |Top 25%| **LLM-CARe (ours)** |84.3/84.5|86.5/86.3|80.4/77.3|
> |25%-50%| LSTM | 72.1/72.1 | 75.5/76.8 | 53.7/49.3 |
> |25%-50%| mtGAN |78.4/78.1|72.1/73.8|56.0/52.2|
> |25%-50%| MedSyn |81.4/81.4|82.3/82.1|72.1/70.0|
> |25%-50%| LLM Direct |79.8/79.7|64.4/65.5|59.1/58.3|
> |25%-50%| **LLM-CARe (ours)** |81.5/81.6|85.1/85.4|74.6/72.8|
> |50%-75%| LSTM | 72.3/72.3 | 74.7/75.6 | 53.7/49.2 |
> |50%-75%| mtGAN |80.3/80.0|71.1/72.2|54.9/50.7|
> |50%-75%| MedSyn |80.2/80.2|82.5/81.9|72.1/70.3|
> |50%-75%| LLM Direct |82.0/81.9|64.2/64.5|57.9/57.0|
> |50%-75%| **LLM-CARe (ours)** |80.9/80.9|84.4/84.5|74.6/73.4|
> |Bottom 25%| LSTM | 70.4/70.4 | 74.7/75.6 | 51.9/47.4 |
> |Bottom 25%| mtGAN |81.1/80.7|71.3/72.9|55.3/51.6|
> |Bottom 25%| MedSyn |81.9/81.9|82.1/81.5|72.6/70.6|
> |Bottom 25%| LLM Direct |83.0/82.8|63.5/65.1|59.9/58.6|
> |Bottom 25%| **LLM-CARe (ours)** |82.5/82.5|84.2/84.6|74.4/72.8|

---

### Official Review · Reviewer_CMTW · 2025-11-01

**Soundness:** 3
**Presentation:** 3
**Contribution:** 3
**Rating:** 4
**Confidence:** 3

**Summary:**

The paper presents LLM-CARe, a large-language-model framework for generating synthetic electronic medical records (EMRs) without using real patient data. It employs three agents—a Critic, Adviser, and Reviser—that work in a cyclic refinement loop to iteratively improve data quality. The process operates in three stages: corpus-level alignment with real-world demographic and diagnostic distributions, section-level completion of clinical fields, and document-level consistency of medical logic. Experiments on a large de-identified EMR dataset show that LLM-CARe surpasses traditional GAN- and LLM-based baselines such as MedSyn in both intrinsic quality metrics and downstream predictive tasks, including diagnosis, examination, and treatment recommendation

**Strengths:**

S1. The authors build an appealing three-agent loop (Critic – Adviser – Reviser) and wrap it in a staged pipeline.

S2. The authors perform extensive evaluations, including both intrinsic “LLM-as-a-judge” metrics and downstream predictive tasks, and even conduct a clinician study to validate alignment between automatic and human assessments.

S3. The experiments are thorough and cover several baselines across different LLM backbones, providing a thorough empirical picture

**Weaknesses:**

W1. The claimed “clinical quality principles” are human-defined checklists, not learned constraints; therefore the method is rule-driven text refinement, not genuine reasoning or data synthesis.

W2. The multi-agent design seems overcomplicated for its marginal gains, with no comparison to a single self-refining LLM.

W3. The approach relies on static, non-temporal EMR data, which limits realism and generalizability to real clinical settings.

W4. The definition of “content completeness” assumes that every section should be fully filled, which may contradict real-world medical documentation patterns where incompleteness reflects diagnostic uncertainty or missing data. Have the authors tested whether enforcing strict completeness might produce clinically implausible or overly templated records?

**Questions:**

Q1. Why three separate agents? Could a single unified agent with self-critique and iterative prompting achieve similar results with lower computational cost?

Q2. Why does the proposed EMR quality not align with downstream task performance. For instance, why does MedSyn, which scores poorly on section-level quality metrics, still achieve competitive results in downstream prediction tasks?

Q3. The document-stage refinement shows limited incremental benefit in Figure 6, raising concerns about its necessity and efficiency. Why is the document-level stage retained when its contribution appears minimal?

---

> ### Author Response · Authors · 2025-11-25
> **Response to Reviewer CMTW**
>
> 1. **Nature of Clinical Quality Principles**: We sincerely appreciate your thoughtful comment on the nature of the clinical quality principles used in our framework. We clarify that our method is not rule-driven in the sense of enforcing fixed templates or surface-level patterns. Instead, the criteria we provide to the agents are high-level clinical requirements, such as “the chief complaint should state the reason for admission,” rather than explicit lexical or structural rules such as “the chief complaint must mention fever.” The multi-agent system evaluates whether a record satisfies these abstract criteria and proposes revisions accordingly, and the final text is synthesized entirely by the LLM rather than being forced into predefined templates.
>
> Importantly, these criteria reflect the same guideline that human clinicians follow when writing and auditing medical records. Clinical documentation in practice is guided by structured principles—for example, what elements belong to each section—and our design mirrors this through the Critic–Adviser–Reviser cycle. The agents reason about whether the content fulfills the expected clinical intent rather than checking for rigid patterns, which allows the system to maintain flexibility and produce diverse, natural records.
>
> Finally, relying solely on implicit constraints learned from data may introduce noise, as real-world EMRs inevitably contain incomplete or inconsistent entries. Using guideline-aligned criteria helps ensure that the generated records satisfy clinically appropriate expectations while still allowing the LLM to freely generate the actual narrative.
>
> We hope this clarifies that LLM-CARe builds on clinically grounded, high-level crireia rather than rule-based text editing, and that the multi-agent refinement process aims to emulate realistic clinical documentation practices.

---

> ### Author Response · Authors · 2025-11-25
> **Response to Reviewer CMTW**
>
> 2. **Comparison with Single Self-Refining LLM**: We are thankful for your constructive remarks regarding the necessity of multiple agents and whether a single unified self-refining model could achieve comparable results. We realize that our original description may not have fully conveyed the structure of our system, which could lead to misunderstanding about the notion of multiple agents. In LLM-CARe, the Critic, Adviser, and Reviser are prompt-defined roles of the same underlying LLM, rather than separate models. This usage follows the established convention in widely adopted multi-agent LLM frameworks such as CAMEL [1] and Reflexion [2], where different prompt-induced behaviors are referred to as distinct “agents” even though they share the same model. In practice, this design does not introduce the architectural overhead of running multiple models; it simply guides the model to think in different modes at different stages.
>
> Beyond this clarification, our framework differs from generic single-agent self-refinement in two key ways that are directly motivated by the characteristics of EMR generation.
>
> First, EMRs must satisfy a large number of high-level clinical requirements, including content completeness, medical correctness, and context consistency. In conventional self-refinement, all requirements are typically presented in a single feedback message, and we found that the model often accommodates only part of these requirements while overlooking others, simply because the combined constraints are too numerous to track in a single revision step. To handle this in a structured way, we organize refinement into three hierarchical stages—corpus, section, and document—and within each stage, we further decompose the checks into fine-grained units, addressing one requirement at a time, enabling focused and reliable refinement across dimensions that interact with each other.
>
> Second, EMR synthesis places constraints not only within each document but also at the corpus level—for example, matching demographic distributions and covering diverse clinical knowledge. These global properties cannot be ensured by existing self-refinement methods that refine each document separately. Instead, LLM-CARe includes a corpus-level stage that explicitly monitors and adjusts population-level patterns, which is essential for generating datasets that support downstream clinical modeling.
>
> To directly evaluate whether a unified self-refining LLM could suffice, we implemented a Self-Refine baseline where a single model critiques and revises drafts using a combined set of section- and document-level criteria. Using the same backbone and producing the same number of EMRs, Self-Refine yields only modest gains over initial drafts and remains far below LLM-CARe across all quality principles. Besides, its downstream task performance is also consistently lower. These results indicate that, in our setting, handling all requirements simultaneously and without corpus-level evaluation is insufficient for achieving high-quality EMRs.
>
> **Quality Score**
>
> | Method          | Content Completeness | Medical Correctness | Context Consistency | Demographic Typicality | Knowledge Coverage |
> | --------------- | ------------ | ----------- | ------------------- | -------------------- | ---------------- |
> | LLM Direct  | 77.1            | 90.7           | 87.9                   | 77.7                    | 73.9                |
> | Self-Refine | 78.3            | 90.9           | 88.5                   | 77.7                    | 78.0                |
> | **LLM-CARe (ours)**    | **91.2**            | **98.6**           | **93.8**                   | **96.8**                    | **94.1**                |
>
> **Downstream Task Performance** (Micro Average/Macro Average)
>
> | Method          | Diagnosis Prediction | Examination Recommendation | Treatment Recommendation |
> | --------------- | --------- | ----------- | --------- |
> | LLM Direct  | 81.8/81.8         | 64.4/65.4           | 60.9/59.0         |
> | Self-Refine | 81.9/81.8         | 64.9/65.7           | 63.1/61.3         |
> | **LLM-CARe (ours)**    | **82.6**/**82.4**         | **85.3**/**85.2**           | **76.9**/**74.1**         |
>
> We hope this clarifies both the terminology and the motivation for our design. Although our framework uses three “agents,” these are different functional roles instantiated by the same model, and the performance differences arise from the structured refinement process rather than architectural complexity.
>
> > [1] Li, G., Hammoud, H., Itani, H., Khizbullin, D., & Ghanem, B. (2023). Camel: Communicative agents for" mind" exploration of large language model society. Advances in Neural Information Processing Systems, 36, 51991-52008.\
> > [2] Shinn, N., Cassano, F., Gopinath, A., Narasimhan, K., & Yao, S. (2023). Reflexion: Language agents with verbal reinforcement learning. Advances in Neural Information Processing Systems, 36, 8634-8652.

---

> ### Author Response · Authors · 2025-11-25
> **Response to Reviewer CMTW**
>
> 3. **Form of EMR Data**: We are grateful for your thoughtful comment regarding EMR format we used. The EMRs used in our study are textual clinical notes that summarize key information from admission, through in-hospital assessment and treatment, to discharge. This form of narrative record is the format commonly adopted in prior work on synthetic textual EMR generation, and our study follows the same data characteristics.
>
> Within this format, the records also contain coarse temporal information that spans different stages of hospitalization. Our document-level refinement incorporates some of these temporal relationships—for example, checking that condition described at admission are consistent with the therapeutic actions recorded later during the stay. While the temporal structure is encoded narratively, the system checks coherence across these stages.
>
> At the same time, we acknowledge that textual EMRs can include much richer forms of temporal structure. These may arise across multiple hospitalizations, across phases of a single hospitalization (e.g., pre-operative, intra-operative, post-operative), or even at finer granularity such as daily progress notes. Modeling such layered temporal structure is an important direction, and we hope to extend our framework to handle these more complex narrative timelines in future work. However, such extensions go beyond the scope of the present study, which focuses specifically on generating narrative clinical notes—the setting most widely explored in existing textual EMR synthesis research. We believe our current formulation is appropriate for this data type, while also leaving room for future expansion toward more temporally detailed textual records.
>
> 4. **Concern about Content Completeness**: We sincerely appreciate your detailed feedback. Our notion of “content completeness” is not a rigid requirement that every field must be fully populated, but rather a set of high-level clinical expectations about what essential information belongs in each section. For example, the chief complaint is expected to indicate the reason for seeking care, but this reason may take many valid forms—an acute symptom, an abnormal test finding, or a follow-up evaluation. The criteria do not prescribe specific wording or specific medical conditions; they only require that the core clinical intent of each section be present.
>
> Completeness is also not enforced in isolation. If a revision that improves completeness introduces inconsistencies or errors, the subsequent document-level refinement will attempt to correct them. The system therefore balances completeness with correctness and consistency rather than prioritizing one dimension at the expense of plausibility.
>
> In addition, real EMRs indeed can contain uncertainty or partially missing information. However, such uncertainty is typically expressed in standard and clinically accepted ways, for example using phrases such as “no obvious trigger” or “diagnosis pending evaluation”. These formulations clearly communicate that the information is unknown without omitting the essential clinical intent of the section. In our framework, such expressions satisfy the completeness criterion, since the requirement is to convey the clinically expected information—not to force unwarranted specificity.
>
> Furthermore, based on our experience conducting extensive EMR quality audits in real clinical settings, we have observed that information omissions often arise from documentation errors, such as physician inexperience, heavy workload, or oversight. These omissions are treated as errors that require correction during clinical quality control, rather than as desirable or meaningful forms of variability. Since our goal is to generate high-quality synthetic EMRs—not to reproduce mistakes that occur in real-world documentation—we consider it appropriate to require the presence of essential information.
>
> We hope this clarifies why the completeness criteria do not lead to unrealistic or overly templated records and instead ensure clinically meaningful content.

---

> ### Author Response · Authors · 2025-11-25
> **Response to Reviewer CMTW**
>
> 5. **Relation between Quality and Task Performance**: Thank you for raising this important question. Downstream predictive tasks are indirect measures of EMR quality: their performance depends on multiple interacting factors beyond any single quality dimension, including task formulation, model capacity, and how well the synthetic data aligns with real data. Therefore, downstream results typically show trend-level alignment with intrinsic quality rather than a strict one-to-one correspondence.
>
> In our experiments, this trend also holds. Although on section-level completeness, MedSyn scores noticeably lower than LLM-CARe, it remains substantially higher than the other baselines. More broadly, across most quality dimensions MedSyn consistently ranks as the second-strongest method. This stronger overall quality explains why MedSyn also demonstrates competitive downstream performance, even though the absolute performance margins across tasks differ.
>
> This pattern supports the expected relationship—quality and downstream utility are aligned at the level of method ranking, though not necessarily in a linear or one-to-one manner—because downstream predictive performance is influenced by multiple interacting aspects of the synthetic data.
>
> 6. **Necessity of Document-Level Refinement**: We are grateful for your thoughtful comments. Document-level refinement targets cross-section consistency and medical correctness, which represent the most clinically consequential types of errors. These issues may be relatively infrequent, but even small inconsistencies can lead to serious implausibilities in a clinical narrative. For example, if the history of present illness mentions a left-hand fracture while the clinical course describes surgery on the right hand, only a few words are incorrect, yet the resulting record is fundamentally flawed. In real EMR quality auditing, such inconsistencies are treated as core errors that must be corrected regardless of their frequency.
>
> In our evaluations, the document-level stage produces clear improvements in these consistency and correctness dimensions, effectively reducing high-impact errors that cannot be addressed by corpus- or section-level refinement alone. It also provides small but consistent gains in downstream diagnosis performance. The numerical improvement is modest because these errors are relatively sparse, yet they are precisely the ones that affect clinical plausibility the most.
>
> For these reasons, we retain the document-level refinement stage: even if its improvement are less significant than other stage, it plays a crucial role in ensuring that the synthetic EMRs meet the clinical coherence and correctness expected of high-quality medical documentation.

---

### Official Review · Reviewer_omri · 2025-11-02

**Soundness:** 3
**Presentation:** 3
**Contribution:** 2
**Rating:** 6
**Confidence:** 3

**Summary:**

This paper proposes LLM-CARe, a  Critic–Adviser–Reviser cyclic refinement framework for synthetic EMR generation. The method refines drafts progressively at corpus, section, and document levels. Each level targets specific quality principles such as distributional alignment, completeness, and consistency. Experiments show that LLM-CARe outperforms baseline GAN- and LLM-based EMR generators on both intrinsic quality metrics (using LLM-as-judge) and downstream clinical tasks (diagnosis/exam/treatment prediction). The paper emphasizes that its method requires no real EMR text, thereby ensuring privacy.

**Strengths:**

* The motivation is clear and well grounded in real-world needs. The paper is clearly written and polished throughout.

* The Critic–Adviser–Reviser framework is intuitive and well explained. The three-stage design (corpus, section, document) aligns nicely with EMR structure and gives the approach a coherent logic.

* Experiments are extensive and consistently favorable. The method improves both intrinsic quality metrics and downstream task accuracy, with convincing ablations and backbone comparisons.

* The inclusion of a clinician study strengthens credibility.

**Weaknesses:**

* The novelty is limited. The multi-agent cyclic refinement closely follows prior self-reflection or debate-style frameworks (e.g. Self-Refine [1]). The contribution is mostly in domain adaptation rather than algorithmic innovation.

* The major intrinsic metrics rely on LLM-based judgments (Qwen-32B), which risks circularity—since the same model family is used for generation and evaluation. Although clinician validation is reported, the sample size (n = 100) is small and does not fully calibrate the quantitative scores in Table 1.

* No comparison to recent diffusion-based EMR generators (e.g., EHRDiff [2])

* No discussion of computational cost or typical failure patterns is provided, leaving scalability and robustness uncertain.

-------
References

[1] Madaan, A., Tandon, N., Gupta, P., Hallinan, S., Gao, L., Wiegreffe, S., ... & Clark, P. (2023). Self-refine: Iterative refinement with self-feedback. Advances in Neural Information Processing Systems, 36, 46534-46594.

[2] Yuan, H., Zhou, S., & Yu, S. EHRDiff: Exploring Realistic EHR Synthesis with Diffusion Models. Transactions on Machine Learning Research.

**Questions:**

* How many iterations per stage were run, and what is the stopping criterion?

* Does the system risk “mode collapse,” where cyclic feedback converges to repetitive templates?

---

> ### Author Response · Authors · 2025-11-25
> **Response to Reviewer omri**
>
> 1. **Novelty Concern**: We sincerely appreciate your thoughtful comments regarding the novelty of our approach. While LLM-CARe is related to iterative self-reflection frameworks, it differs in two essential ways that are important for generating clinically valid EMRs.
>
> **First**, generic self-refinement methods typically present all quality requirements and all detected issues in a single feedback message, and the model is expected to revise the entire document in one step. In EMR generation, however, the number of requirements is very large and some of them may influence one another. When many requirements are given at once, we generally observe that the model fails to fully follow some of them or only applies partial corrections, simply because the instruction contains too many constraints to track simultaneously. In contrast, LLM-CARe checks one requirement at a time and applies a targeted correction only for that requirement, allowing the model to satisfy quality criteria more reliably.
>
> **Second**, most self-refinement approaches operate at the level of a single document, while synthetic EMR generation also requires that the entire dataset follow realistic demographic and knowledge distributions. LLM-CARe introduces a corpus-level critic to adjust population-level statistics, a capability that is important for downstream clinical models and is generally not included in refinement methods that focus on a single document at a time.
>
> To further address this concern, we implemented a Self-Refine [1] baseline: all section- and document-level criteria were combined into one feedback instruction, and each draft was evaluated and revised accordingly. Using the same backbone and generating the same number of EMRs, we evaluated both intrinsic quality and downstream task performance. The results are summarized below.
>
> **Quality Score**
> | Method          | Content Completeness | Medical Correctness | Context Consistency | Demographic Typicality | Knowledge Coverage |
> | --------------- | ------------ | ----------- | ------------------- | -------------------- | ---------------- |
> | LLM Direct  | 77.1            | 90.7           | 87.9                   | 77.7                    | 73.9                |
> | Self-Refine | 78.3            | 90.9           | 88.5                   | 77.7                    | 78.0                |
> | **LLM-CARe (ours)**    | **91.2**            | **98.6**           | **93.8**                   | **96.8**                    | **94.1**                |
>
> **Downstream Task Performance** (Micro Average/Macro Average)
> | Method          | Diagnosis Prediction | Examination Recommendation | Treatment Recommendation |
> | --------------- | --------- | ----------- | --------- |
> | LLM Direct  | 81.8/81.8         | 64.4/65.4           | 60.9/59.0         |
> | Self-Refine | 81.9/81.8         | 64.9/65.7           | 63.1/61.3         |
> | **LLM-CARe (ours)**    | **82.6**/**82.4**         | **85.3**/**85.2**           | **76.9**/**74.1**         |
>
> Across quality principles, Self-Refine yields slightly higher section-level and document-level scores (content completeness, medical correctness, and context consistency) than LLM Direct, but it remains clearly below LLM-CARe across all these dimensions. This indicates that asking the model to handle a long list of requirements in a single revision step is not effective, and that decomposing the refinement into separate, requirement-specific steps is necessary to achieve reliable improvements. For corpus-level quality (demographic typicality and knowledge coverage), Self-Refine shows only a small improvement over LLM Direct and is still far behind LLM-CARe. This demonstrates that without an explicit corpus-level evaluation and adjustment mechanism, single-document refinement cannot ensure that the synthetic dataset follows realistic demographic and knowledge distributions—an essential property for EMR generation. Besides, Self-Refine also performs worse than LLM-CARe on all downstream tasks, due to low quality of the generated EMRs.
>
> Overall, these findings show that compared with generic iterative refinement methods, LLM-CARe’s cyclic refinement design—separating feedback across quality principles and incorporating corpus-level constraints—is essential for generating clinically meaningful synthetic EMRs.
>
> > [1] Madaan, A., Tandon, N., Gupta, P., Hallinan, S., Gao, L., Wiegreffe, S., ... & Clark, P. (2023). Self-refine: Iterative refinement with self-feedback. Advances in Neural Information Processing Systems, 36, 46534-46594.

---

> ### Author Response · Authors · 2025-11-25
> **Response to Reviewer omri**
>
> 2. **LLM-as-Judge Choice & Human Evaluation Scale**: We are sincerely grateful for your constructive comments. We have taken several experiments to address both the potential circularity in LLM-based evaluation and the limited size of the clinician evaluation.
>
> To reduce dependence on a single model family for evaluation, we conducted additional assessment with two independent judges. GPT-OSS-20B was selected because its strong performance on medical tasks, and DeepSeek-V3.2-Exp (API) was included as a larger commercial model to further test robustness. The DeepSeek evaluation was conducted on 10% sampled data for cost consideration.
>
> The results are shown in the tables below. The overall trends remain consistent: LLM-CARe achieves the top scores under all three evaluators and maintains a clear margin over the baselines, whereas the relative ordering among the other methods shows only minor differences across models. These observations suggest that our conclusions do not depend on the choice of a specific judge and that the superiority of LLM-CARe is robust across evaluation settings.
>
> **Qwen2.5-32B Evaluation**
> | Method      | Content Completeness | Medical Correctness | Context Consistency |
> | ----------- | ------------ | ----------- | ----------- |
> | LSTM        | 70.8            | 65.0           | 21.7           |
> | mtGAN       | 55.8            | 51.8           | 21.4           |
> | MedSyn      | 84.8            | 95.3           | 91.9           |
> | LLM Direct  | 77.1            | 90.7           | 87.9           |
> | **LLM-CARe**    | **91.2**            | **98.6**           | **93.8**           |
>
> **GPT-OSS-20B Evaluation**
> | Method      | Content Completeness | Medical Correctness | Context Consistency |
> | ----------- | ------------ | ----------- | ----------- |
> | LSTM        | 73.9            | 88.9           | 8.9           |
> | mtGAN       | 62.8            | 87.7           | 13.4           |
> | MedSyn      | 79.9            | 97.3           | 86.9           |
> | LLM Direct  | 81.1            | 96.7           | 70.2           |
> | **LLM-CARe**    | **93.4**            | **99.0**           | **91.3**           |
>
> **DeepSeek-V3.2-Exp Evaluation**
> | Method      | Content Completeness | Medical Correctness | Context Consistency |
> | ----------- | ------------ | ----------- | ----------- |
> | LSTM        | 69.9            | 72.5           | 2.6           |
> | mtGAN       | 57.9            | 64.0           | 0.7           |
> | MedSyn      | 80.6            | 97.5           | 85.0           |
> | LLM Direct  | 79.9            | 96.8           | 77.8           |
> | **LLM-CARe**    | **91.1**            | **98.5**           | **89.8**            |
>
> To assess the stability of human–LLM agreement, we extended the clinician evaluation beyond the original 100 samples. Clinicians annotated two additional sets of 50 EMRs each, sampled to maintain balanced coverage across generation methods, gender, age groups, and both common and less-common diseases. We then recomputed agreement metrics using subsets of 100, 150, and 200 annotated samples. The agreement values and their confidence intervals remain similar across these three sample sizes, indicating that the 100-sample evaluation already provides a reasonably stable estimate of human alignment.
>
> | # Samples | Clinician-LLM Agreement (Cohen’s κ) | 95% CI (Cohen’s κ) |  Inter-Clinician Agreement (Fleiss’s κ) | 95% CI (Fleiss’s κ) |
> |----------|-----------|-------------|----------|--------------|
> | 100      | 0.837         | [0.804, 0.868]           | 0.947        | [0.928, 0.964]            |
> | 150      | 0.842         | [0.816， 0.868]           | 0.948        | [0.933, 0.962]            |
> | 200      | 0.833         | [0.809, 0.855]           | 0.950        | [0.937, 0.962]            |
>
> Together, these analyses demonstrate that our intrinsic quality assessment is stable, supporting the reliability of our evaluation protocol.

---

> ### Author Response · Authors · 2025-11-25
> **Response to Reviewer omri**
>
> 3. **Comparison with Diffusion-Based EMR Generation**: Thank you very much for pointing us to this line of work and for suggesting a comparison. We carefully reviewed the referenced diffusion-based method, which focuses on generating structured numerical EHR representations (e.g., patient features encoded as binary vectors). In response to your suggestion, we attempted to extend diffusion-based generation to our setting by adopting a diffusion-LM [2] approach to synthesize textual EMRs.
>
> However, despite extensive effort, we were unable to obtain meaningful textual outputs: the model consistently produced incoherent character sequences rather than valid clinical language. We experimented with commonly used stabilization techniques—including logit scaling and EMA updates—but these adjustments did not resolve the issue. Our analysis suggests that several domain-specific factors likely contribute to the failure mode, such as the long-text nature of clinical notes, the relatively limited dataset size available for diffusion training, and the prevalence of specialized medical terminology not well supported by pretrained word embedding.
>
> We sincerely appreciate the reviewer’s suggestion. We are continuing to experiment with diffusion-based approaches, and if we obtain successful results, we will update our response accordingly. We will include a discussion in the revised manuscript comparing diffusion-based EMR generators with our framework in terms of data modality, applicability, and potential advantages.
>
> > [2] Li, X., Thickstun, J., Gulrajani, I., Liang, P. S., & Hashimoto, T. B. (2022). Diffusion-lm improves controllable text generation. Advances in neural information processing systems, 35, 4328-4343.
>
> 4. **Computational Cost and Failure Patterns**: We would like to thank the reviewer for the thoughtful feedback regarding scalability and robustness. To address this point, we provide a detailed analysis of both computational cost and typical failure patterns observed in our system.
>
> Regarding computational cost, LLM-CARe requires additional refinement steps compared with direct generation. On a single RTX 4090 GPU, generating 38k synthetic EMRs takes approximately 13 hours with direct generation, while LLM-CARe requires around 36 hours, including both initial draft generation and the three-stage refinement process. Although our framework performs three levels of Critic–Adviser–Reviser interactions, the overhead is mitigated by two factors: (1) the outputs of these agents are significantly shorter than full EMRs, and (2) drafts that pass the critic at any stage do not proceed to subsequent agents, avoiding unnecessary refinement cycles. Importantly, the computational cost grows linearly with the number of generated records, and the refinement steps can be parallelized across disease categories, ensuring practical scalability to larger corpora.
>
> To better characterize model robustness, we also examined typical failure patterns in the generated EMRs. One typical issue involves inconsistent onset times when multiple symptoms are present. As shown in Example 1, when the chief complaint specifies different durations for fever, chills, and cough, the generated history of present illness may occasionally assign mismatched durations to these symptoms, leading to temporal inconsistency. Another recurrent pattern is incomplete documentation of general conditions. Clinical practice typically requires recording appetite, urination, bowel movements, mental status, sleep, and weight change in the history of present illness. However, as shown in Example 2, synthetic records sometimes include only a subset of these elements, resulting in reduced completeness. These cases illustrate that current large language models still face challenges when multiple clinical attributes must be simultaneously coordinated—such as aligning several symptom timelines or ensuring full coverage of multiple required general-condition items. Future work may address these limitations through further task decomposition or more fine-grained refinement strategies.
>
> > Example 1 — Inconsistent Symptom Onset Times \
> > Chief Complaint: Fever for 3 days; **chills and cough for 1 day** \
> > History of Present Illness: The patient developed fever 3 days ago with a peak temperature of 39.5°C, accompanied by **chills for 3 days**. One day prior to admission, the patient also developed a cough…
> >
> > Example 2 — Incomplete Documentation of General Conditions \
> > History of Present Illness: The patient developed fever 2 days ago, accompanied by chills and general malaise... The patient reports reduced appetite, poor mental energy, and inadequate sleep.

---

> ### Author Response · Authors · 2025-11-25
> **Response to Reviewer omri**
>
> 5. **Iteration Count and Stopping Criterion**: Thank you for your detailed feedback regarding the iterative procedure. In our framework, the stopping criterion is applied individually to each EMR, rather than at the corpus level. At every stage, the Critic evaluates the draft against the requirements specific to that stage (corpus-, section-, or document-level). If a draft already satisfies all requirements, it does not undergo further refinement within that stage and proceeds directly to the next stage. This per-record early stopping ensures that unnecessary iterations are avoided.
>
> For robustness, we also set an upper bound of two refinement rounds per stage. Empirically, we observed that additional iterations provide minimal improvement while increasing computational cost. To validate this observation, we conducted an ablation by increasing the maximum number of iterations to three rounds per stage. As shown in tables below, the results remain nearly unchanged compared with the two-round setting. These findings indicate that, under our current setup, the refinement process already converges within two iterations, and additional rounds do not yield meaningful gains.
>
> **Quality Score**
>
> | Maximum Iterations per Stage          | Content Completeness | Medical Correctness | Context Consistency | Demographic Typicality | Knowledge Coverage |
> | --------------- | ------------ | ----------- | ------------------- | -------------------- | ---------------- |
> | 2    | 91.2            | 98.6           | 93.8                   | 96.8                    | 94.1                |
> | 3    | 91.1            | 98.1           | 93.8                   | 96.8                    | 95.6                |
>
> **Downstream Task Performance** (Micro Average/Macro Average)
>
> | Maximum Iterations per Stage          | Diagnosis Prediction | Examination Recommendation | Treatment Recommendation |
> | --------------- | --------- | ----------- | --------- |
> | 2    | 82.6/82.4         | 85.3/85.2           | 76.9/74.1         |
> | 3    | 82.9/82.9         | 85.7/85.5           | 76.7/73.9         |
>
> 6. **Risk of Mode Collapse**: We sincerely appreciate this thoughtful question. Ensuring that cyclic refinement does not reduce diversity is an important concern, and we address it from both a design perspective and an empirical perspective.
>
> From a methodological standpoint, LLM-CARe explicitly incorporates corpus-level constraints that require the synthetic EMRs to reflect the distributional breadth of real clinical data. During corpus-level alignment, the system is encouraged to cover the range of clinical knowledge and demographics observed in the real corpus, rather than collapsing onto only a small number of “typical” cases. At the section- and document-level stages, the refinements do not prescribe stylistic structure or fixed templates, and therefore do not force records into repetitive narrative formats.
>
> In practice, we found that initial drafts generated directly by LLMs are more prone to repetition, since each record is produced independently without considering corpus-level coverage. After applying LLM-CARe’s staged refinement, the resulting EMRs exhibit better variability. To quantify this, we follow mtGAN [3] and compute the Self-BLEU score across the synthetic corpus, where lower values indicate higher diversity. As shown in the table below, the refined EMRs obtained through LLM-CARe achieve lower Self-BLEU compared with the initial drafts from direct generation, demonstrating that our framework increases diversity and mitigates the risk of mode collapse.
>
> |Method|Self-BLEU(↓)|
> |-|-|
> |EMRs before cyclic refinement|0.962|
> |**EMRs after cyclic refinement**|**0.859**|
>
> > [3] Guan, J., Li, R., Yu, S., & Zhang, X. (2019). A method for generating synthetic electronic medical record text. IEEE/ACM transactions on computational biology and bioinformatics, 18(1), 173-182.

---

> ### Author Response · Authors · 2025-12-01
> **Response to Reviewer omri**
>
> **Results on Diffusion-Based EMR Generation**
>
> We are grateful for your thoughtful comments. To provide a direct comparison with diffusion-based EMR generation, we employed LLaDA-MoE-7B-A1B-Instruct [4], a recent diffusion language model, to synthesize textual EMRs from prompts. As shown in the tables below, the EMRs produced by the diffusion model achieve substantially lower intrinsic quality and downstream task performance—even underperforming the LLM Direct baseline. These results indicate that current diffusion models may still be insufficient for generating complex clinical narratives that must satisfy multiple quality requirements, which explains why existing diffusion-based EMR generation (e.g. EHRDiff) focused predominantly on numeric EMR data rather than textual records.
>
> **Quality Score**
> | Method          | Content Completeness | Medical Correctness | Context Consistency | Demographic Typicality | Knowledge Coverage |
> | --------------- | ------------ | ----------- | ------------------- | -------------------- | ---------------- |
> | Diffusion Model (LLaDA)  | 69.3            | 95.8           | 35.5                   | 71.9                    | 36.6                |
> | LLM Direct  | 77.1            | 90.7           | 87.9                   | 77.7                    | 73.9                |
> | **LLM-CARe (ours)**    | **91.2**            | **98.6**           | **93.8**                   | **96.8**                    | **94.1**                |
>
> **Downstream Task Performance** (Micro Average/Macro Average)
> | Method          | Diagnosis Prediction | Examination Recommendation | Treatment Recommendation |
> | --------------- | --------- | ----------- | --------- |
> | Diffusion Model (LLaDA)  | 79.8/79.9         | 55.7/56.9           | 46.0/47.9         |
> | LLM Direct  | 81.8/81.8         | 64.4/65.4           | 60.9/59.0         |
> | **LLM-CARe (ours)**    | **82.6**/**82.4**         | **85.3**/**85.2**           | **76.9**/**74.1**         |
>
> > [4] Zhu, F., You, Z., Xing, Y., Huang, Z., Liu, L., Zhuang, Y., ... & Wen, J. R. (2025). LLaDA-MoE: A Sparse MoE Diffusion Language Model. arXiv preprint arXiv:2509.24389.

---

### Comment · Area_Chair_TYNm · 2025-11-28
**A gentle reminder to participate in the author–reviewer discussion.**

Dear Reviewers,

Thank you once again for your service to ICLR 2026. Now that the authors have submitted their rebuttal, could you please engage in the interactive discussion with them? Your participation would be very helpful to the authors, and they would greatly appreciate it. Please also read the authors’ response together with the other reviews and consider whether the rebuttal or any additional comments influence your assessment of the paper.

Thank you again for your efforts.

Best wishes,

Your AC

---

### Author Response · Authors · 2025-11-29
**General Responses**

We thank all reviewers for their thoughtful comments and constructive suggestions. We appreciate the recognition of the LLM-CARe design, which applies stage-wise Critic–Adviser–Reviser cyclic refinement to synthesize high-quality EMRs, leading to consistent improvements in both intrinsic EMR quality and downstream task performance, all achieved without training on or prompting with real EMR text.

Based on the reviewers’ valuable feedback, we have made the following key additions and clarifications:

1. Comparison and Analysis with More Baseline and Variant
    - Comparison with Self-Refine Baseline: Following the reviewers’ suggestions, we added a generic self-refinement baseline. Results show that generic self-refinement offers only limited improvements, underscoring the importance of our structured multi-agent refinement design. (see lines 317-319, Table 1-2, Figure 5-6)
    - Analysis of an LLM-CARe Variant: We further analyzed a variant of our framework that incorporates real EMR text during refinement. Results show that whether real EMR text is included has only minimal influence on performance, indicating that the gains primarily arise from the structured refinement process itself. (see Section 5.6, Figure 9)
2. Issues of Evaluation
    - LLM Evaluation Issue: Based on the reviewers' suggestions,we further include two additional model families (GPT-OSS and DeepSeek) for intrinsic EMR quality evaluation, and the results show that LLM-CARe consistently achieves the highest performance across all evaluators. (see lines 321-323, Appendix F)
    - Downstream Task Issue: To make the downstream evaluation more rigorous, we supplemented the experiments with an external dataset (MIMIC) and an additional backbone model (Llama). Results show that LLM-CARe consistently improves performance across all settings, demonstrating that downstream gains are robust across datasets and model architectures. (see lines 385-386, Appendix H)
    - Issue of Clinician Evaluation: To validate the effectiveness of LLM-based evaluation, the original paper has included a clinician evaluation conducted by four licensed clinicians. Following the reviewers’ suggestions, we engaged the same clinicians and doubled the annotation size to provide a more rigorous assessment. In the original setting, both clinician–LLM agreement and inter-clinician agreement were already high; after doubling the annotations, these agreements remained highly stable, demonstrating that our initial clinician evaluation was already reliable. (see lines 501-503, Appendix I)
3. Clarifications of Reviewers' Concerns
    - Clarification of Related Work: Besides the original related work on GAN-, autoregressive-, and LLM-based EMR generation, we followed the reviewers’ suggestions to further include a recent LLM-based study and clarify that current diffusion-based methods primarily target numeric EMRs thus not suitable for our textual EMR generation setting. (see lines 116, 118-120)
    - Clarification of Quality Principles: In addition to the full description already provided in the appendix, we enhanced the main-text presentation by adding an illustrative figure with representative criteria under each quality principle, making the principles more intuitive and easier to understand. (see lines 128-137, Figure 2)
    - Clarification of Baseline Methods: Following the reviewers’ suggestions, we further provide more implementation details of baseline methods. (see lines 270-279, Appendix C)
    - Clarification of Corpus-Level Reference Distribution: Given the reviewers’ suggestions, we clarified that the reference distributions are aggregated statistics (gender, age, entity frequencies) derived from the training set. (see lines 182-183)
    - Clarification of Less Typical Cases: Based on the reviewers' suggestion, we refine the phrasing to clarify that our focus is on improving coverage of lower-frequency but clinically valid conditions within a disease category. (see lines 46-47)
    - Clarification of the Relationship Between Quality and Downstream Tasks: We further clarified that while downstream task performance does not correspond directly to any single quality dimension, the overall trend is consistent: methods achieving higher intrinsic EMR quality also tend to perform better on downstream predictive tasks. (see lines 379-381)

---

### Author Response · Authors · 2025-11-29
**General Responses**

Although large language models possess strong generation capabilities and are promising tools for EMR synthesis, our practical experience shows that existing approaches—whether prompting with real EMR exemplars or applying generic refinement strategies—face fundamental limitations. These include mismatched corpus-level distributions, incomplete section content, and logical inconsistencies across document fields, all of which prevent current synthetic EMR methods from meeting the standards required for real-world clinical use.

Through long-term collaboration with hospital medical record departments in developing EMR quality-control systems, we observed that real EMRs are written and audited according to explicit, well-established quality principles. Motivated by this insight, we developed LLM-CARe, a stage-wise multi-agent cyclic refinement framework operating at the corpus, section, and document levels. At each level, refinement is guided by the corresponding quality principles, and the three agents—a Critic that evaluates drafts against these principles, an Adviser that provides targeted revision suggestions, and a Reviser that updates the records—collaborate in a cyclic loop to enhance quality. Evaluations demonstrate that LLM-CARe substantially improves corpus-level distributional realism, section-level content completeness, and document-level logical consistency—all without relying on real EMR text—and that the resulting synthetic EMRs offer superior utility for downstream clinical tasks.

Overall, producing synthetic EMRs that satisfy stringent clinical quality principles while remaining broadly useful is a challenging yet essential goal. We believe that our framework, which integrates clinical quality principles with structured multi-agent refinement, provides a meaningful step toward bringing synthetic EMR generation closer to real-world application.

---

### Author Response · Authors · 2025-12-03
**Rebuttal Summary for the New Area Chair**

Dear Area Chair,

We sincerely appreciate your time and effort in supporting the review process under the current circumstances. Recognizing the many demands on your time, we have prepared a concise summary of our work and the key interactions during the rebuttal phase to help streamline your assessment.

Our work focuses on synthetic EMR generation, an increasingly important direction for enabling privacy-preserving clinical research and supporting downstream medical AI development. Although modern large language models possess strong generative capabilities, we find that directly applying them to EMR synthesis—either by prompting with real EMRs or using generic refinement strategies—introduces several issues. These include distributional mismatch at the corpus level, incomplete section-level content, and logical inconsistencies across document fields. Consequently, current synthetic EMR methods fail to produce records that meet the quality standards required for real-world clinical use.

To address these challenges, we propose LLM-CARe, a stage-wise multi-agent cyclic refinement framework operating at the corpus, section, and document levels. Each stage is explicitly guided by corresponding clinical quality principles, and three agents—a Critic that evaluates drafts against quality principles, an Adviser that formulates targeted feedback, and a Reviser that updates the record—collaborate in a structured refinement loop. Our evaluations show that LLM-CARe markedly improves corpus-level distributional alignment, section-level content completeness, and document-level logical consistency—all without accessing real EMR text. Furthermore, the resulting synthetic records provide stronger utility for downstream clinical tasks, demonstrating the practical value of our framework.

During the review process, several reviewers highlighted the clinically grounded motivation for high-quality EMR synthesis (omri, UAQV, 3q4A), the structured stage-wise Critic–Adviser–Reviser refinement design (omri, CMTW, 3q4A), the extensive evaluation spanning both intrinsic quality and downstream tasks (omri, CMTW, UAQV, 3q4A), and the reliability of the findings supported by ablation studies, backbone comparisons, and clinician evaluation (omri, CMTW, UAQV, 3q4A).

Based on valuable feedback from the reviewers, we have systematically addressed the concerns. These include further comparisons with a generic self-refinement baseline and variants of our framework, confirming that the improvements arise from the proposed stage-wise multi-agent refinement; incorporating additional model families and an external dataset for evaluation, as well as doubling the clinician evaluation size to strengthen rigor; and clarifying presentation details, such as providing a more explicit explanation of the corpus-level reference distributions.

Overall, ensuring that synthetic EMRs satisfy rigorous clinical quality principles while retaining practical utility is challenging yet essential for real-world clinical use. By grounding generation in explicit quality requirements and enforcing them through a stage-wise multi-agent refinement process, our framework offers a meaningful step toward synthetic EMRs that can more effectively support real-world applications.

Thank you again for your time and thoughtful consideration of our submission. We hope our work contributes to the advancement of high-quality and clinically applicable EMR generation systems.

Best regards,

Authors of Submission 16898

---

### Meta-Review · Area_Chair_XBff · 2026-01-06

**Summary:**

This paper studies synthetic EMR generation with LLMs under privacy constraints.

The core method is a refinement piepline to operate at various granuarities each with multiple roles to polish.

Across the reviewers, the motivation is clear anc critical. The major cocnerns center around the novelty and evaluation circularity by using LLMs as a judge,etc.

**Reviewer Concerns:**

Novelty remains the main limitation: the approach is largely a structured composition of known prompting patterns -- whcih we know as critique and revision loops. Of course, there is something new like enforce contraints at different granularities. In short, it is more engineering and empirical.

Second, the framework relies on human-specified “quality principles.” Even if they are high-level, the method is still driven by a checklist-like specification, instead of a learned constraint model.

Generalization may still be questionable given the data setting is mostly discharge style notes.

Finally, the computational cost is non-retrial. "On a single RTX 4090 GPU, producing 38k synthetic EMRs takes approximately 13 hours
with direct generation, whereas LLM-CARe requires about 36 hours". Maybe it is fine.

**Reviewer Scores:**

Only Reviewer 3q4A (8) responded and likely the score will stay the same.

Reviewer omri (6) in my view may stay the same or even bump to 8 given the initial concerns of novelty, evaluation, and others are reasonably addressed in my view.

Reviewer UAQV (4) -- I think  nearly all concerns were explicitly addressed. The rebuttal is unusually thorough. I am confident a 6 will be given.

Reviewer CMTW (4) may still stay unchanged.

given a 8864 ir 8664 score, I believe the paper is in the publishable state.

---

### Decision · Program_Chairs · 2026-01-26

Accept (Poster)